# Improving lysosomal ferroptosis with NMN administration protects against heart failure

Mikako Yagi[1,2], Yura Do[1], Haruka Hirai[1,2], Kenji Miki[1], Takahiro Toshima[1], Yukina Fukahori[1], Daiki Setoyama[1], Chiaki Abe[3], Yo-Ichi Nabeshima[3], Dongchon Kang[1], Takeshi Uchiumi[1,2]

**Myocardial mitochondria are primary sites of myocardial energy metabolism. Mitochondrial disorders are associated with various cardiac diseases. We previously showed that mice with cardiomyocyte-specific knockout of the mitochondrial translation factor p32 developed heart failure from dilated cardiomyopathy. Mitochondrial translation defects cause not only mitochondrial dysfunction but also decreased nicotinamide adenine dinucleotide (NAD+) levels, leading to impaired lysosomal acidification and autophagy. In this study, we investigated whether nicotinamide mononucleotide (NMN) administration, which compensates for decreased NAD+ levels, improves heart failure because of mitochondrial dysfunction. NMN administration reduced damaged lysosomes and improved autophagy, thereby reducing heart failure and extending the lifespan in p32cKO mice. We found that lysosomal damage due to mitochondrial dysfunction induced ferroptosis, involving the accumulation of iron in lysosomes and lipid peroxide. The ameliorative effects of NMN supplementation were found to strongly affect lysosomal function rather than mitochondrial function, particularly lysosome-mediated ferroptosis. NMN supplementation can improve lysosomal, rather than mitochondrial, function and prevent chronic heart failure.**

## Introduction

Mitochondria are important organelles that produce ATP and are recognized as metabolic hubs because they interact with various organelles in the cell and play essential roles in vital processes (Anderson et al, 2019). Membrane contact sites between mitochondria and other organelles, such as the endoplasmic reticulum and lysosomes, are important for signaling, lipid exchange, and membrane dynamics (Murley & Nunnari, 2016). The existence of mitochondria–endoplasmic reticulum contact sites is well known (Dorn, 2013), and their involvement in fatty acid metabolism,

calcium signaling pathways, autophagy, and inflammasomes has also been reported (Pinton, 2018; Yang et al, 2020).

In recent years, the interaction between mitochondria and lysosomes has been considered essential for cellular homeostasis. Lysosomes have V-ATPase, which keeps the inside of lysosomes acidic, and fuses with autophagosome membranes and participates in autophagic degradation mechanisms. Therefore, lysosomes maintain homeostasis in living organisms by degrading and reusing unwanted intracellular substances. Specifically, lysosomes contribute to intracellular energy production by generating amino acids, sugars, and lipids (Todkar et al, 2017).

A relationship between lysosomes and mitochondria is also found in the regulation of their morphology. Lysosomes are involved in mitochondrial fission, and mitochondria regulate lysosomal dynamics (Miyazaki & Toumon, 2019). Dysfunction of one of these functions is thought to affect the other, leading to various pathological conditions or exacerbation of pathological states such as neurodegenerative diseases (e.g., Parkinson's disease, Alzheimer's disease, lysosomal storage disease, and amyotrophic lateral sclerosis) (Osellame & Duchen, 2014). Mitochondrial dysfunction leads to impaired lysosomes and autophagy, and functional and morphological mitochondrial dysfunction may be essential for the development of these lysosomal pathologies (Audano et al, 2018).

The *p32/C1qbp* gene functions as an essential RNA and protein chaperone in mitochondrial translation and is indispensable for embryonic development (Yagi et al, 2012). We previously found that the mitochondrial translation regulator p32 is involved in various diseases, such as leukoencephalopathy and cardiomyopathy, in a mouse model (Saito et al, 2017; Yagi et al, 2017). In particular, mice with cardiomyocyte-specific knockout of p32 (p32cKO) also showed mitochondrial dysfunction and a shortened lifespan of ~1 yr. One of the causes of this shortened lifespan was increased hypoxia-inducible factor 1 (HIF-1)-α expression and decreased nicotinamide adenine dinucleotide (NAD+) levels and lysosomal dysfunction. We also reported that ATP required for lysosomal activity was generated from NAD+ via the glycolytic enzymes GAPDH and

[1]Department of Clinical Chemistry and Laboratory Medicine, Graduate School of Medical Sciences, Kyushu University, Fukuoka, Japan  [2]Department of Health Sciences, Graduate School of Medical Sciences, Kyushu University, Fukuoka, Japan  [3]Department of Aging Science and Medicine, Graduate School of Medicine Kyoto University Medical Innovation Center, Kyoto, Japan

Correspondence: uchiumi.takeshi.008@m.med.kyushu-u.ac.jp

phosphoglycerate kinase. We propose that local ATP production is required for lysosomal activation and have suggested a novel mechanism for the generation of ATP from NAD$^+$ on the lysosomal outer membrane (Yagi et al, 2021).

Various metabolic processes in vivo involve organelle communication (Hönscher et al, 2014). Lysosomes are regulated by mitochondria, which affect NAD$^+$/NADH levels and reactive oxygen species production (Baixauli et al, 2015; Demers-Lamarche et al, 2016). NAD$^+$ is mainly synthesized via nicotinamide mononucleotide (NMN). Therefore, NMN administration has been reported to increase NAD$^+$ (Yoshino et al, 2018). Nicotinamide phosphoribosyl transferase (NAMPT) plays a major role in the adipose-to-brain signaling pathway that sustains the association between increased adiposity and impaired sociability triggered by early life stress. In adipose tissue–specific normalization of NAMPT levels, dietary treatment with NMN in adolescent-stressed mice normalizes the alterations in sociability and neuronal excitability through the activation of the NAD$^+$/sirtuin 1 (SIRT1) pathway (Morató et al, 2022). Furthermore, oral supplementation with nicotinamide riboside, a vitamin B3 form and NAD$^+$ precursor, efficiently prevents the development and progression of mitochondrial myopathy in mice (Khan et al, 2014).

NAD$^+$ levels in the hearts of mice protects against ischemia–reperfusion injury (Yamamoto et al, 2014; Yoshino et al, 2018). Several reports have implicated Sirt1 (NAD$^+$-dependent deacetylase) as the mechanism of the NMN effect. In one report, ischemic preconditioning activated Sirt1, which induced deacetylation of lysine residues in proteins, and regulated protein post-translational modifications by maintaining availability of the substrate NAD$^+$ (Nadtochiy et al, 2011). Supplementation with the precursor NMN to increase NAD$^+$ levels improved cardiac function in several disease mouse models (Martin et al, 2017; Wu et al, 2021).

When systemic NMN production is reduced with aging, certain tissues respond first with a decrease in Sirt1 activity and associated metabolic changes. Furthermore, Sirt1 has been reported to regulate cell metabolism and lysosomal acidification in cancer cells (McAndrews et al, 2019). NMN administration improves lysosomal function, and the detailed mechanism is related to an improvement in ferroptosis in lysosomes.

Ferroptosis is an iron-dependent cell death reported to involve mitochondria (Gao et al, 2019; Gan, 2021) and is induced by the intracellular accumulation of lipid peroxide, glutathione peroxidase 4 (GPX4) inactivation, and low glutathione (GSH) levels generated from cysteine. Mitochondria are involved in the generation of lipid peroxide, but the detailed mechanism is unknown (Mao et al, 2021). The expression of the ferroptosis-related genes prostaglandin-endoperoxide synthase 2 (Ptgs2), acyl CoA-synthetase long-chain family member 4 (Acsl4), and ChaC glutathione specific gamma-glutamylcyclotransferase 1 (Chac1) has been reported to be up-regulated by ferroptosis (Mungrue et al, 2009; Wang et al, 2019; Chen et al, 2021). Lysosomal function is essential for intracellular iron metabolism because free intracellular iron is toxic and iron is thought to be stored in lysosomes (Rizzollo et al, 2021). Therefore, we investigated whether mitochondrial dysfunction induces ferroptosis in lysosomes. We then examined whether NMN administration improves lysosomal and autophagic function and prevents ferroptosis-induced heart failure in the p32cKO model.

# Results

## Cardiac function improves with NMN administration

Many preclinical cardiomyopathy mouse models use ischemia–reperfusion injury, and NMN is often provided by injection for only a few weeks and observed for a short period. In this study, we investigated whether long-term administration of NMN restores cardiac function and extends the lifespan. NMN was added to the drinking water of mice (1 mg/ml), and WT and p32cKO mice were fed NMN from 2 to 9 mo of age to examine its effect on cardiac function. We observed that mRNA expression of atrial natriuretic factor (ANF) and β-myosin heavy chain (β-MHC) (heart failure markers) was increased in p32cKO mice, but this was partially ameliorated by NMN treatment (Fig 1A). p32cKO promoted myocardial fibrosis, which was also reversed by NMN administration (Fig 1B).

In patients with congestive heart failure and in experimental animal models of heart failure, dopamine and epinephrine have been reported to be increased in the myocardium (Zausig et al, 2010). Epinephrine is secreted mainly from the medulla of the adrenal glands and is released in response to acute stress. Therefore, we analyzed metabolites of dopamine and epinephrine, and found that they were significantly increased in p32cKO myocardium, but were improved by NMN administration (Fig 1C). We previously reported that mice with cardiomyocyte-specific p32KO had a short lifespan (Saito et al, 2017). We found that treatment of these mice with NMN significantly prolonged their lifespan (Fig 1D). However, NMN treatment did not improve the heart size or morphology (Fig S1A). These results suggest that NMN treatment functionally ameliorated or prevented exacerbation of heart failure in p32 cardiomyocyte–specific knockout mice, thereby prolonging their lifespan.

## Improvement of myocardial NAD$^+$ with NMN administration

We then used metabolomic analysis to determine whether NMN administration restores the amount of NAD$^+$ in myocardial tissue. NAD$^+$ was decreased in p32cKO myocardium and recovered to levels similar to those in the WT by NMN administration (Fig 2A). In addition, nicotinamide (NAM) was significantly increased in WT and p32cKO myocardium by NMN administration. This finding indicated that myocardial NAD$^+$ levels were restored by NMN administration. However, NADH (NAD, reduced form) and NADP$^+$ (NADP, oxidized form) were decreased in p32cKO myocardium, but NMN administration had no effect on them (Fig 2A).

Levels of mRNA expression of members of the NAD$^+$-mediated salvage pathway, such as Nampt, Nmnat1, Nmnat3, and Sirt3, were up-regulated by NMN administration, which indicated that the amount of NAD$^+$ in vivo was increased (Fig 2B). We previously reported that NAD$^+$ synthase Nmnat3 expression was regulated by the transcription factor HIF-1α (Yagi et al, 2021). Therefore, we investigated the effect of NMN treatment on HIF-1α protein levels in p32cKO mice (Fig 2C). HIF-1α protein levels was increased in p32cKO mice and restored by NMN administration. This finding may have been due to restoration of HIF-1α protein levels, which in turn restored Nmnat3 gene expression.

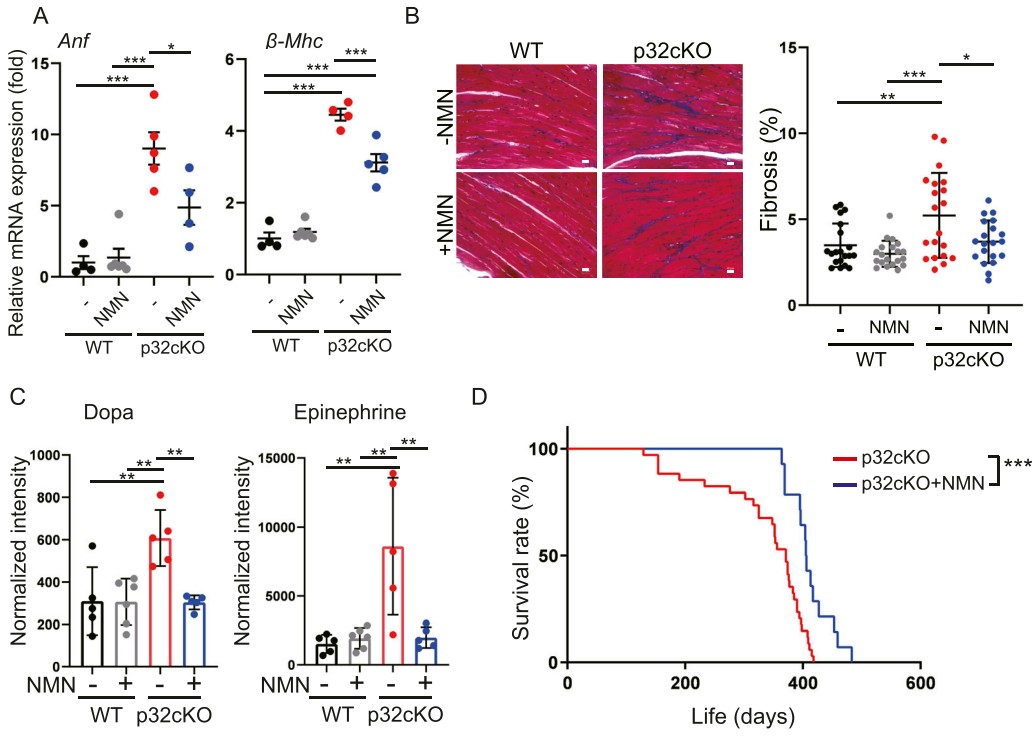

**Figure 1. Nicotinamide mononucleotide (NMN) administration improves cardiac dysfunction in p32cKO mice.**
**(A)** Relative mRNA expression levels of the cardiomyopathy biomarkers atrial natriuretic factor and β-myosin heavy chain in the heart. WT and p32cKO mice were treated with or without NMN from 2 to 9 mo of age (WT: n = 4, WT + NMN: n = 6, p32cKO: n = 5, p32cKO + NMN: n = 4). ***$P$ < 0.0001, *$P$ = 0.0302. **(B)** Cardiac fibrosis as shown by Masson's trichrome staining for WT and p32cKO mice treated from 2 to 9 mo of age with NMN (n = 3). Eight images were taken for each mouse and the proportions of blue-stained areas were plotted (20 locations in total). Quantification is shown in the right panel. Scale bars, 20 μm. ***$P$ = 0.0001, **$P$ = 0.0043, *$P$ = 0.0152. **(C)** Metabolite analysis of dopamine and epinephrine in the heart of WT and p32cKO mice treated with NMN from 2 to 9 mo of age (WT, p32cKO, and p32cKO + NMN: n = 5, WT + NMN: n = 6). **(D)** Kaplan–Meier survival curve for p32cKO mice treated with or without NMN (p32cKO: n = 34, p32cKO+NMN: n = 14) (***$P$ = 0.0005) were analyzed by the log-rank test (**$P$ = 0.0016). **(A, B, C)** Error bars show the mean ± SD. Data were analyzed by one-way analysis of variance with Tukey's multiple comparisons test. *$P$ < 0.05, **$P$ < 0.01, ***$P$ < 0.001.

## Restoration of lysosomal function by NMN administration

We previously reported that lysosomal function was impaired owing to decreased NAD⁺ biosynthesis in vivo (Yagi et al, 2021). Because NAD⁺ biosynthesis was increased with NMN administration (Fig 2A), we investigated whether lysosomal function is also restored by NMN administration. Levels of mRNA expression of the transcription factor *Tfeb*, which regulates lysosomal biosynthesis (Settembre et al, 2012), were decreased in p32cKO and recovered by NMN administration (Fig 3A). These results suggest that NMN-mediated *Tfeb* expression improves lysosomal function.

Lysosomes break down unwanted intracellular substances and convert them into amino acids for metabolite recycling and storage, to maintain homeostasis of intracellular nutritional status. In this study, comparative amounts of amino acid in heart tissue were measured by metabolomic analysis. We found that glutamine and histidine levels were decreased and asparagine levels were increased in p32cKO hearts, and that NMN administration increased these three amino acids (Fig 3B). These three amino acids are dependent on V-ATPase, which is essential for lysosomal activity (Abu-Remaileh et al, 2017). Other amino acids were unchanged between WT and p32cKO hearts, but all amino acid levels were increased by NMN administration (Fig S1B). This finding suggests

that NMN improves lysosomal function and activates amino acid metabolism. Lipofuscin is an insoluble fluorescent pigment that autofluorescence from lipid peroxides that accumulate with age in the lysosomal compartment. Therefore, as lysosomal function declines, lipofuscin accumulates within lysosomes and can cause disease (Pierzynowska et al, 2021). We observed that the accumulation of lipofuscin around the nucleus was increased by p32cKO and recovered by NMN administration (Fig 3C). These results suggest that NMN administration improves lysosomal function in the p32cKO heart.

We then confirmed lysosomal morphology by electron microscopy. Lysosomes were larger in the p32cKO heart, which was thought to be an indicator of lysosomal dysfunction, but this increase was not improved by NMN administration (Fig S1C). These results suggest that NMN administration improves lysosomal function, but not lysosomal morphology, in the p32cKO heart.

## Restoration of autophagic function by NMN administration

Protein levels of Lamp2, which is a lysosomal marker protein, were up-regulated in p32cKO compared with those in the WT but decreased and recovered to WT levels with NMN administration (Fig 4A). Immunostaining for Lamp2 showed that it was accumulated

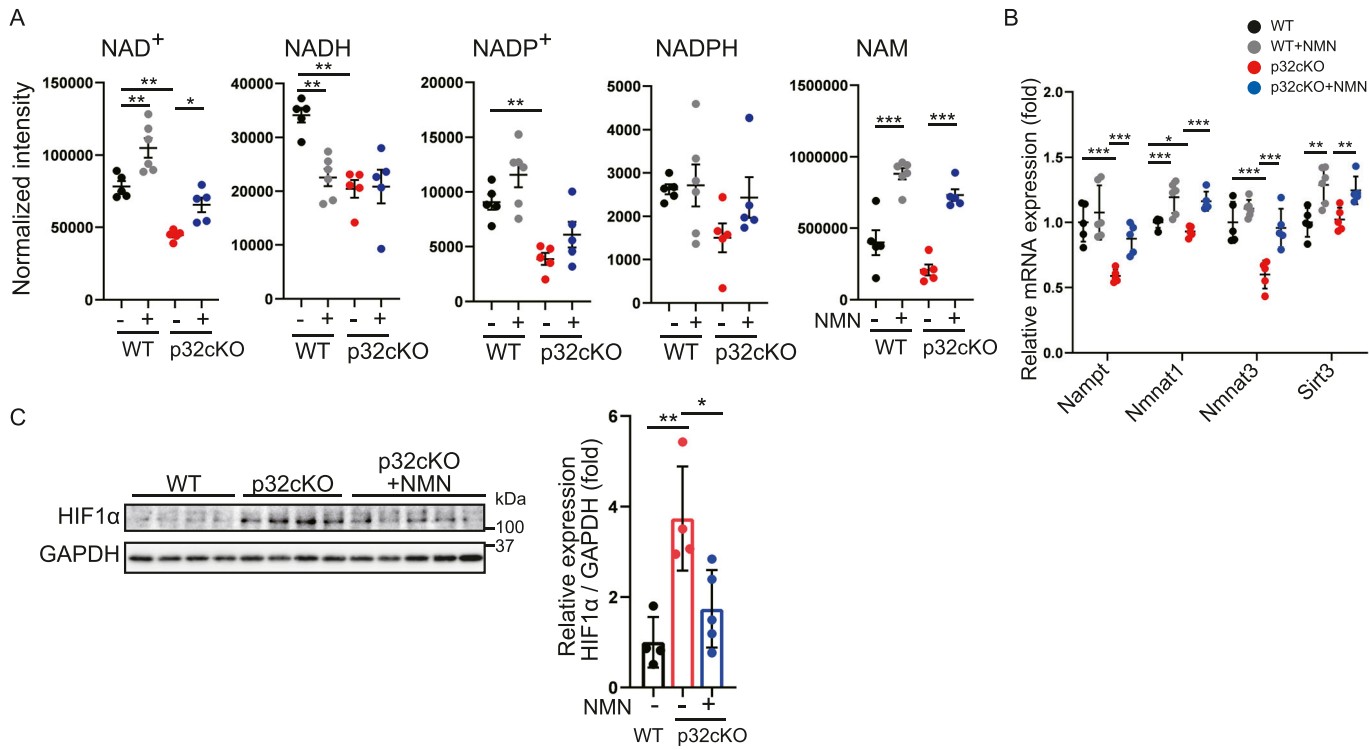

**Figure 2. Nicotinamide mononucleotide (NMN) administration restores NAD⁺ levels in the p32cKO heart.**
**(A)** LC-MS/MS metabolomic analysis of nicotinamide adenine dinucleotide (NAD⁺), nicotinamide (NAD⁺), nicotinamide adenine dinucleotide, reduced form (NADH), nicotinamide adenine dinucleotide monophosphate (NADP⁺), and nicotinamide adenine dinucleotide monophosphate reduced form (NADPH) in heart tissue of 9-mo-old WT and p32cKO mice treated with or without NMN from 2 to 9 mo of age (WT, p32cKO, and p32cKO + NMN: n = 5, WT + NMN: n = 6). NAD⁺ and NADP⁺ show significantly different levels between WT and p32cKO hearts. Data are presented as the mean ± SEM. **(B)** RNA expression of NAD⁺-synthesizing enzymes in WT and p32cKO hearts under treatment with or without NMN from 2 to 9 mo of age (WT: n = 5, WT + NMN: n = 6, p32cKO and p32cKO + NMN: n = 5). Data are presented as the mean ± SD. **(C)** Western blot analysis of hypoxia-inducible factor 1α in the heart of WT and p32cKO mice, and of p32cKO mice treated with NMN from 2 to 9 mo of age. Quantification is shown in the right panel (WT and p32cKO: n = 4, p32cKO + NMN: n = 5). **$P = 0.0037$, *$P = 0.0181$. GAPDH was used as an internal control. Error bars show the mean ± SD. **(A, B, C)** Data were analyzed by one-way analysis of variance with Tukey's multiple comparisons test. *$P < 0.05$, **$P < 0.01$, ***$P < 0.001$.
Source data are available for this figure.

mostly around the nucleus in p32cKO mice (Fig S1D), which suggested lysosomal dysfunction because lamp2-staining patterns exhibit a morphology similar to lipofuscin. To further investigate lysosomal function, we performed immunostaining for galectin-3, which has been reported to accumulate in damaged lysosomes (Jia et al, 2020). The P32cKO mice showed increased galectin-3 staining and colocalization of Lamp2 and galectin-3, which suggested that lysosomes were damaged, but damaged lysosomes were partially restored by NMN administration in p32cKO hearts (Fig 4B).

Next, we also investigated whether autophagic function was restored by NMN administration. Protein levels of the autophagy marker p62 were higher in p32cKO mice than in WT mice (Fig 4C). Furthermore, NMN treatment improved p62 protein levels to WT levels (Fig 4C). Ring morphology was confirmed by p62 immunostaining of myocardial tissue, in which the autophagosome membrane was not degraded by autophagy. Quantification of the ring morphology showed that it was more common in p32cKO than in WT mice and tended to improve with NMN administration (Fig 4D). NMN administration also restored the function of lysosomes and thus improved the function of autophagy, which suggested that NMN administration improved the degradation mechanism of autophagy.

## NMN administration does not improve mitochondrial function and morphology

We have reported that p32cKO causes mitochondrial translation defects, decreases the expression of CoxI encoded by mitochondrial DNA, and decreases oxidative phosphorylation (Yagi et al, 2012). In this study, we further examined whether NMN administration restores mitochondrial function of the p32cKO heart. Comparison of CoxI protein levels in cardiac tissues of WT, p32cKO, and p32cKO treated with NMN mice showed no improvement with NMN treatment (Fig 5A). Mitochondrial morphology was observed by electron microscopy, and internally collapsed giant mitochondria were more commonly observed in the p32cKO heart than in the WT heart. However, NMN treatment partially reduced the number of abnormal mitochondria (Fig 5B). Metabolomic analysis of components of the tricarboxylic acid cycle (TCA) showed that isocitrate, malate, and pyruvate levels were reduced in p32cKO mice compared with their levels in WT mice (Fig S2A). Only 2-ketoglutamic acid levels were increased by NMN addition in WT and p32cKO mice. 2-Ketoglutamic acid is closely related not only to the TCA cycle but also to the glutamine metabolic reaction. Gene expression of *GLUD1*, which is the enzyme that produces 2-ketoglutamic acid from

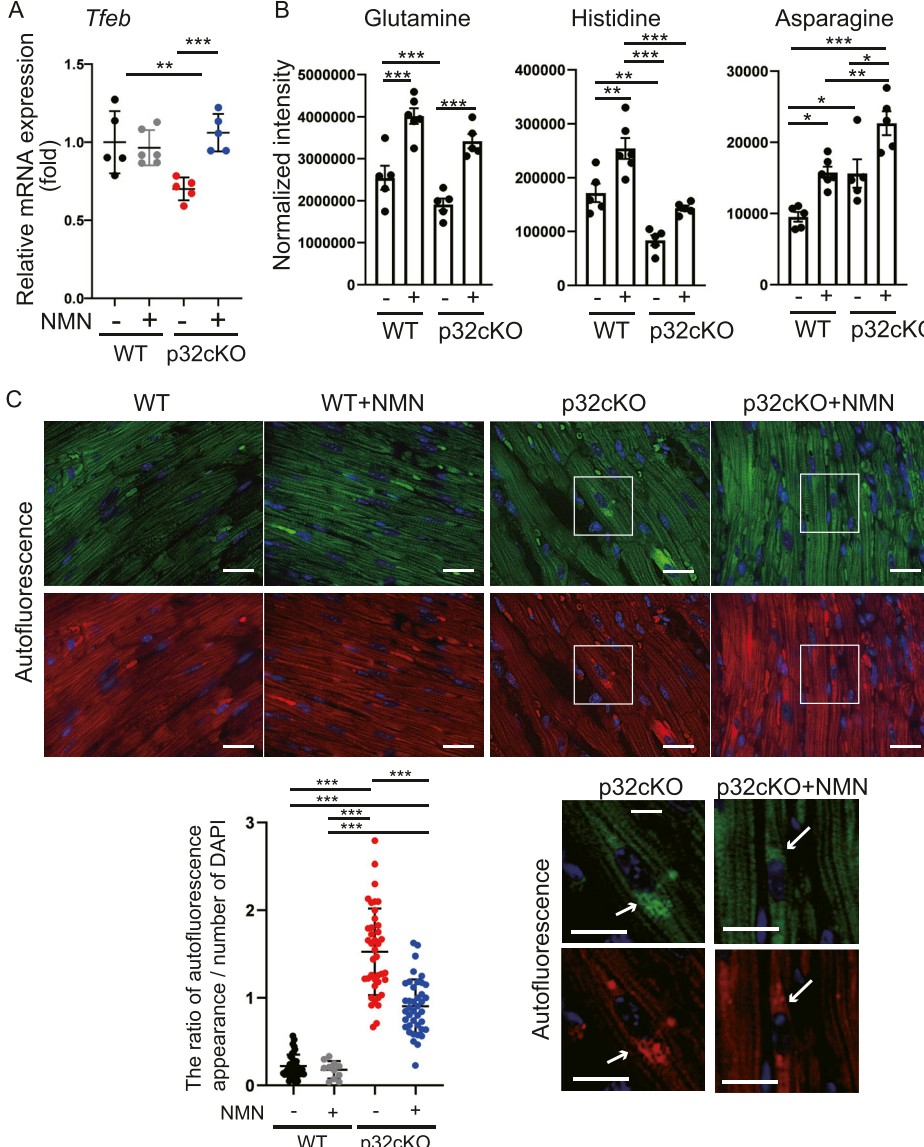

**Figure 3. Nicotinamide mononucleotide (NMN) administration restores lysosomal function in the p32cKO heart.**
**(A)** RNA expression in the heart of WT and p32cKO mice subjected to NMN treatment from 2 to 9 mo of age (WT, p32cKO, and p32cKO + NMN: n = 5, WT + NMN: n = 6). Data are presented as the mean ± SD. **(B)** Amino acid metabolism analysis of NMN-treated mice. LC-MS/MS metabolomic analysis of amino acids was performed in 9-mo-old WT and p32cKO mice and mice supplemented with NMN from 2 to 9 mo of age (WT, p32cKO, and p32cKO + NMN: n = 5, WT + NMN: n = 6). Amino acids are significantly different between p32cKO and p32cKO + NMN. Data are presented as the mean ± SEM. **(C)** Autofluorescence showing lipofuscin localization around the nucleus as dots per cell in the heart of WT and p32cKO mice, and in p32cKO mice treated with NMN from 2 to 10 mo of age. Tissues were excited at a wavelength of 540 nm (upper panel) or 470 nm (lower panel), and emission spectra were collected with a confocal microscope at wavelengths (band path) of 580–630 nm (right panel) or 510–560 nm (left panel). Scale bars, 20 μm. Quantification of the ratio of autofluorescent blots per DAPI-staining area is shown in the lower panel as the mean ± SD (n = 4). Arrows indicate intracellular aggregation of lipofuscin.

glutamine, was also up-regulated only with the addition of NMN (Figs 3B and S2A), which suggested that NMN administration improved glutamine metabolism. However, no ameliorating effect of NMN on the overall TCA cycle was observed in p32cKO mice.

We previously reported that reduced oxygen consumption and altered expression of genes associated with mitochondrial function were observed in p32cKO heart tissue (Saito et al, 2017). In this study, we observed that reduced basal and maximal respiration was not recovered by NMN administration (Figs 5C and S2B). We found mRNA expression of *Fgf21* and *Gdf15*, which are markers of mitochondrial disease (Saito et al, 2017), but could not show any improvement with NMN administration (Fig 5D). Therefore, NMN administration did not restore functional mitochondrial damage caused by the inhibition of mitochondrial translation. These findings suggest that NMN administration improves heart failure by improving lysosomal function without improving mitochondrial function.

## Heart-specific mitochondrial translation defect involves ferroptosis

We next examined iron metabolism and ferroptosis, which are closely related to lysosomal function. The suppression of ferroptosis ameliorates heart failure (Fang et al, 2019; Chen et al, 2021). We found that *Ptgs2* and *Chac1* mRNA expression levels were higher in p32cKO mice than in WT mice, which indicated that ferroptosis was induced in p32cKO myocardial tissue (Figs 6A and S3A). Furthermore, NMN administration suppressed only *Chac1* mRNA expression in p32cKO mice. Therefore, we investigated the ferroptosis-related factors affected by *Chac1* mRNA expression. *Chac1* gene expression is regulated by the transcription factor ATF4, which is altered by mitochondrial stress (Wang et al, 2019; Kitakata et al, 2021). Previous reports have shown increased mitochondrial stress and transcription factor *ATF4* expression in

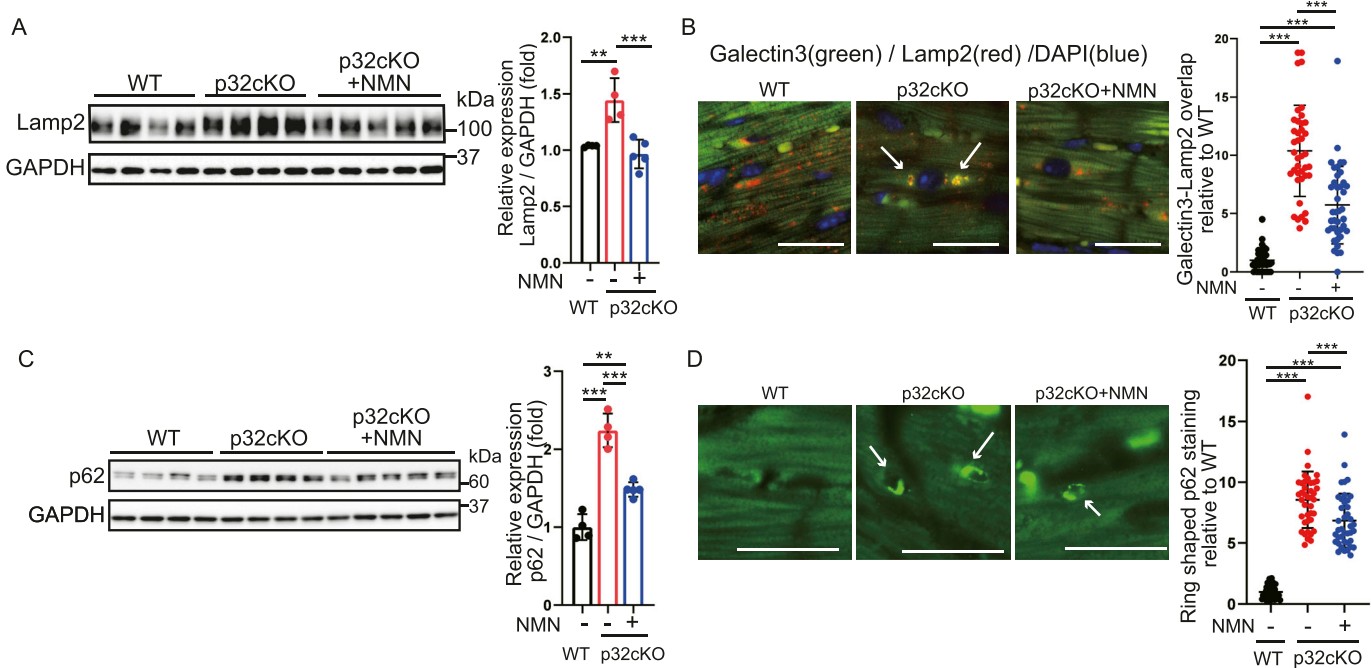

**Figure 4. Nicotinamide mononucleotide (NMN) administration enables recovery from autophagic degradation in the p32cKO heart.**
**(A)** Western blot analysis of Lamp2 in the heart of WT and p32cKO mice and of NMN-treated p32cKO mice. Quantification is shown in the right panel (WT and p32cKO: n = 4, p32cKO + NMN: n = 5). **$P = 0.0039$, ***$P = 0.0009$. Data are presented as the mean ± SD. GAPDH was used as an internal control. **(B)** Immunostaining of galectin-3 (green), Lamp2 (red), and DAPI (blue) in the heart tissue of WT and p32cKO mice, and in p32cKO mice treated with NMN from 2 to 9 mo. The right panel shows the levels of galectin-3 and Lamp2 overlap per cell corrected with DAPI (n = 4). Data are presented as the mean ± SD. Scale bars, 20 $\mu m$. Arrows indicate the sites of colocalization of galectin-3 and Lamp2. **(C)** Western blot analysis of the autophagy marker protein p62 in the heart in 9-mo-old WT and p32cKO mice, and in p32cKO mice treated with NMN from 2 to 9 mo of age. GAPDH was used as an internal control. Quantification is shown in the right panel (WT and p32cKO: n = 4, p32cKO + NMN: n = 5). **$P = 0.0031$, ***$P < 0.0001$. Values are presented as the mean ± SD. **(D)** Immunostaining of p62 (green) in heart tissue of WT and p32cKO mice and of p32cKO mice treated with NMN from 2 to 9 mo of age. The magnified image shows the ring-like staining pattern in the p32cKO heart. The right panel shows the number of ring-shaped formations per cell as the mean ± SD (n = 4). Scale bar, 20 $\mu m$. Arrows indicate p62 ring formation. Data were analyzed by one-way analysis of variance with Tukey's multiple comparisons test. *$P < 0.05$, **$P < 0.01$, ***$P < 0.001$.
Source data are available for this figure.

p32cKO mice (Saito et al, 2017; Sasaki et al, 2017, 2020). In our study, *Atf4* mRNA expression was increased in mitochondrial translational dysfunction and was improved by the addition of NMN (Fig 6B). The mRNA expression of *Chop*, which is an endoplasmic reticulum stress marker, were increased in p32cKO mice and restored by the addition of NMN (Fig 6C). CHOP is also thought to be involved in ATF4–CHOP–CHAC1 pathway-induced ferroptosis (Mungrue et al, 2009; Wang et al, 2019). In addition, protein levels of GPX4, which is the main regulator of ferroptosis, were up-regulated in p32cKO mice and restored by NMN administration, which suggested that NMN improved ferroptosis (Fig 6D).

Chac1 is involved in degradation of GSH and overexpression of Chac1 reduced GSH level, and it was observed that the amount of GSH and oxidized GSH (GSSG), which are substrates of GPX4, was reduced (Fig 6E). This finding suggests that a reduction in GSH may reduce GPX4 activity, leading to the promotion of ferroptosis. The mRNA expression of the ferroptosis-associated factor heme oxygenase-1 (*Ho1*), which promotes the release of iron from heme, was increased in p32cKO mice and improved with NMN administration (Fig 6F). Transcription factor, nuclear factor erythroid 2–related factor 2 (Nrf2) has been reported to be associated with mitochondrial function, and its expression is thought to be up-regulated by a p32 defect (Kasai et al, 2020; Miki

et al, 2022). In addition, mRNA expression of the *Ho1* and transcription factor *Nrf2* was increased in p32cKO mice and improved by NMN administration (Fig 6G).

Lipid peroxide (4-HNE, 3-nitrotyrosine) in the heart was examined. Protein levels and tissue staining showed that lipid peroxide was accumulated intracellularly in p32cKO mice and improved by NMN administration in all experiments (Figs 6H and I and S3B and C). The ferroptosis-related factors Chac1, GPX4, and Ho1 were also ameliorated by NMN (Fig 6J), which suggested that ferroptosis in the p32cKO heart was improved by NMN administration. To investigate ferroptosis in more detail, intracellular dynamics of divalent iron (Fe2+) and lipid peroxidation were examined in cultured cells.

### Ferroptosis resulting from mitochondrial dysfunction is improved by NMN

Ferroptosis involves iron deposition and iron-induced lipid peroxides. We examined in vitro whether iron is deposited in lysosomes and lipid peroxide accumulates in p32 knockdown cells, and whether this is ameliorated by NMN administration. Cytosolic iron levels were quantified in p32 knockdown HeLa cells using FerroOrange reagent, which detects cytosolic iron (Fe$^{2+}$) in living cells. The addition of erastin, a ferroptosis inducer, increased the

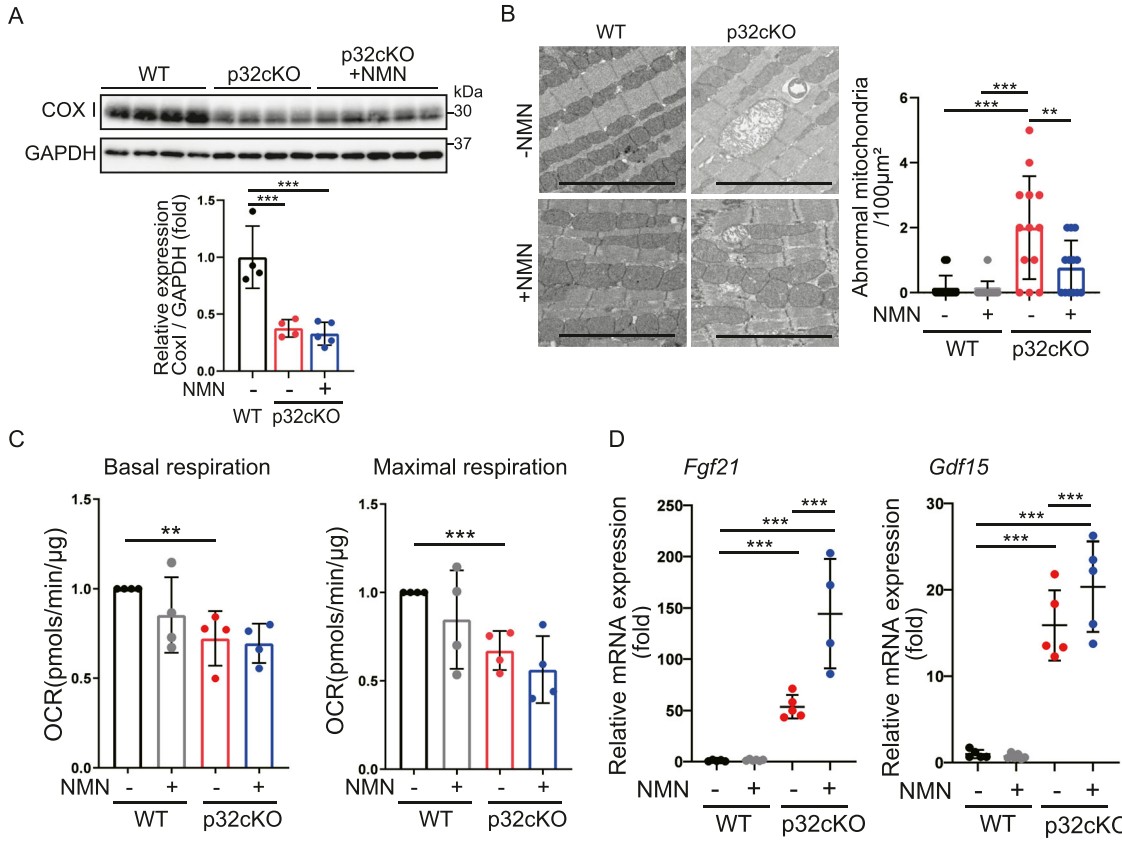

**Figure 5. Mitochondrial dysfunction is not rescued by nicotinamide mononucleotide (NMN) administration.**
**(A)** Western blot analysis of the mitochondrial DNA-encoded protein CoxI in 9-mo-old WT and p32cKO mice, and in p32cKO mice treated with NMN from 2 to 9 mo of age. GAPDH was used as an internal control. Quantification is shown in a graph. Values are presented as the mean ± SD (WT and p32cKO: n = 4, p32cKO + NMN: n = 5). **(B)** Mitochondrial morphology in the hearts of 9-mo-old WT and p32cKO mice, and of WT and p32cKO mice treated with NMN from 2 to 9 mo of age observed by electron microscopy. Scale bars, 5 $\mu$m. Quantification of the average area of abnormal mitochondria is shown in a graph, and data are shown as the mean ± SD (n = 3). **(C)** The oxygen consumption rate in heart mitochondria was analyzed with the flux analyzer. Basal respiration and maximal respiration were calculated as respiratory parameters. WT and p32cKO mice were treated with or without NMN from 2 to 9 mo of age. Values are presented as the mean ± SD (n = 4). **(D)** Real-time polymerase chain reaction analysis of mRNA expression in the heart of WT and p32cKO mice treated with NMN from 2 to 9 mo of age. The mitochondrial disease biomarkers *Fgf21* (fibroblast growth factor 21) and *Gdf15* (growth and differentiation factor 15) are shown. Data are presented as the mean ± SD (n = 5). Data were analyzed by one-way analysis of variance with Tukey's multiple comparisons test. *$P < 0.05$, **$P < 0.01$, ***$P < 0.001$.
Source data are available for this figure.

intensity of cytosolic iron (Fe$^{2+}$) ions and lipid peroxides (Fig S4A and B). The intensity of cytoplasmic ferrous ion was higher in p32 knockdown cells than in WT cells, but decreased when treated with NMN (Fig 7A). These results suggested that ferroptosis was induced in the p32 knockdown cells and restored by NMN treatment. The lysosomal markers LysoPrime Green and FerroOrange were co-localized in p32 knockdown cells (Fig 7B); therefore, most of the intracellularly accumulated (Fe$^{2+}$) was localized to lysosomes. We then investigated whether lipid peroxides were increased in p32 knockdown cells, and quantified this by determining the intensity of LipiRADICAL Green. We observed that p32 knockdown cells had higher lipid peroxide levels that WT cells, whereas NMN treatment decreased lipid peroxide levels (Fig 7C). Furthermore, LipiRADICAL Green and LysoTracker Red were co-localized in p32 knockdown cells (Fig 7D), which suggested the accumulation of lipid peroxides in lysosomes. These results suggested that in p32 knockdown cells, lysosomal function was impaired, lipid peroxide accumulation in lysosomes

was increased, and ferroptosis was induced, but NMN treatment restored lysosomal function and ameliorated ferroptosis. The amount of divalent iron in mitochondria was quantified using MitoFerroGreen and was not different between WT and p32 knockdown cells (Fig 7E). This finding suggested that p32 dysfunction did not alter the amount of Fe$^{2+}$ in mitochondria, but it altered Fe$^{2+}$ in lysosomes.

We then focused on the amount of NAD$^+$ in iron-related lipid peroxides and examined the relationship between the amount of NAD$^+$ and ferroptosis because NAD$^+$ is reduced in p32cKO mice. When intracellular NAD$^+$ levels were lowered using the Nampt inhibitor FK866, intracellular iron deposition, and lipid peroxide were induced and then improved by NMN treatment (Fig 7F and G). These results suggest that the induction of ferroptosis in lysosome is closely related to the amount of NAD$^+$ biosynthesis. Indeed, mitochondrial translation defects were observed in p32 knockdown cells and intracellular NAD$^+$ and NADH levels were reduced, confirming the induction of ferroptosis (Fig S4C).

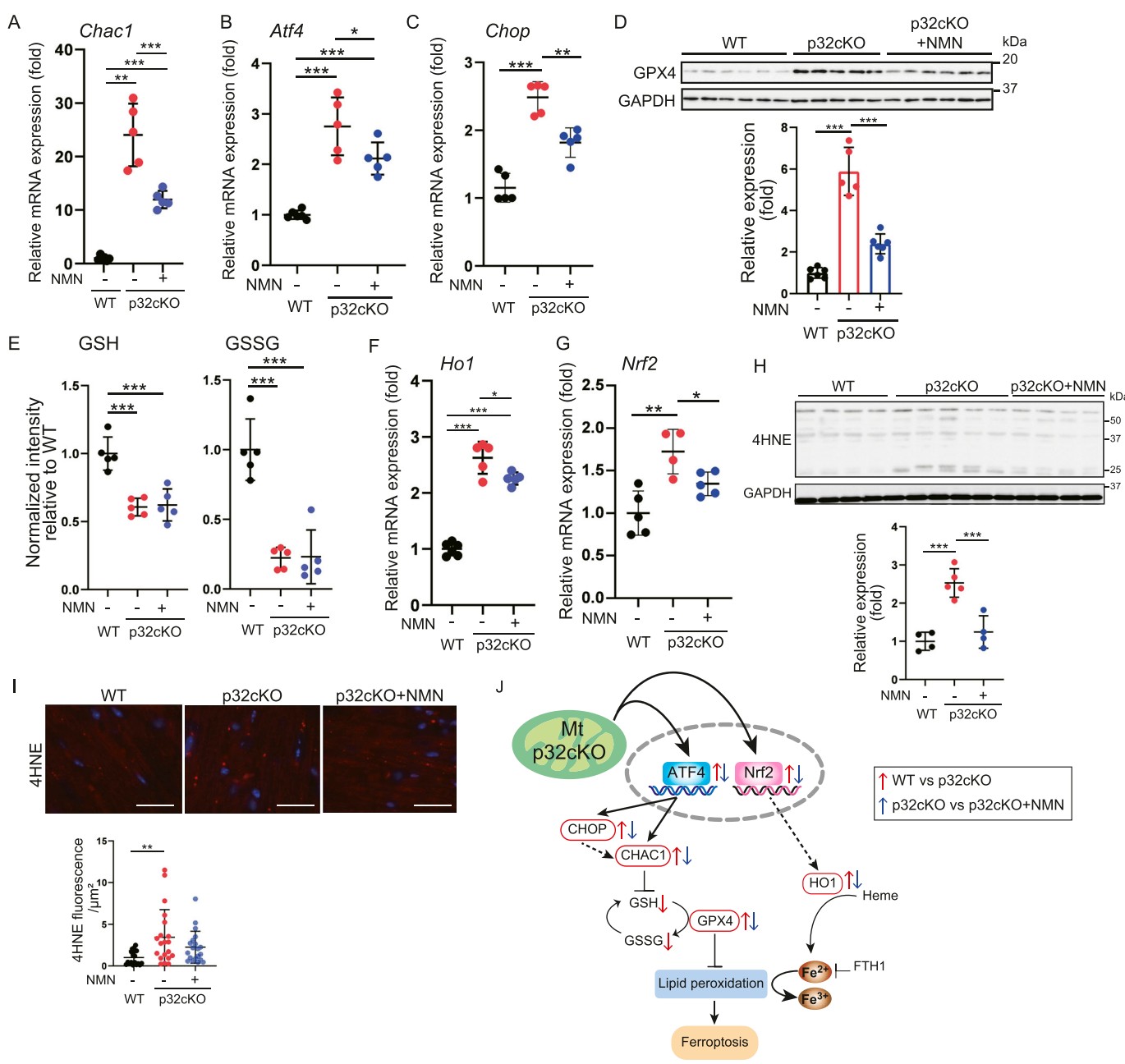

**Figure 6. Mitochondrial translation deficiency induces ferroptosis.**

**(A, B, C)** Real-time polymerase chain reaction analysis of RNA expression in the heart of WT and p32cKO mice treated with nicotinamide mononucleotide (NMN) from 2 to 11 mo of age (WT, p32cKO, and p32cKO + NMN: n = 5). *Chac1* was used as a marker of ferroptosis and an ATF4 inducer. *Atf4* and *Chop* were used as markers of mitochondrial stress. Data are presented as the mean ± SD. **(D)** Western blot analysis of the ferroptosis-related protein glutathione peroxidase 4 in the heart of 11-mo-old WT and p32cKO mice, and of p32cKO mice treated with NMN from 2 to 11 mo of age (WT and p32cKO + NMN: n = 6, p32cKO: n = 5). GAPDH was used as an internal control. Quantification is shown in a graph. Values are presented as the mean ± SD. **(E)** LC-MS/MS metabolomic analysis of GSH and GSSG in 9-mo-old WT and p32cKO mice, and in mice treated with NMN from 2 to 9 mo of age (WT, p32cKO, and p32cKO + NMN: n = 5). Data are presented as the mean ± SEM. **(F, G)** Real-time polymerase chain reaction analysis of mRNA expression in the heart of WT and p32cKO mice treated with NMN from 2 to 11 mo of age (WT, p32cKO, and p32cKO + NMN: n = 5). Data are presented as the mean ± SD. **(H)** Western blot analysis of lipid peroxidation (shown by 4-HNE) to assess various ferroptosis markers in the heart of 11-mo-old WT and p32cKO mice and of p32cKO mice treated with NMN from 2 to 11 mo of age (WT and p32cKO + NMN: n = 4, p32cKO: n = 5). GAPDH was used as an internal control. Quantification is shown in a graph. Values are presented as the mean ± SD. **(I)** Immunostaining of 4-HNE (red) and DAPI (blue) in the heart tissue of WT and p32cKO mice and of p32cKO mice treated with NMN from 2 to 11 mo. The right panel shows the level of 4-HNE overlap per cell corrected with DAPI (n = 4). Data are presented as the mean ± SD. Scale bars, 20 µm. **(A, B, C, D, E, F, G, H, I)** Data were analyzed by one-way analysis of variance with Tukey's multiple comparisons test. *$P < 0.05$, **$P < 0.01$, ***$P < 0.001$. **(J)** The ferroptosis pathway is changed in the p32cKO heart. Red arrows indicate an increase or decrease in the p32cKO heart compared with the level in the WT heart. Blue arrows indicate the comparison between p32cKO and p32cKO + NMN. Red squares indicate genes of which expression was restored by the addition of NMN. Source data are available for this figure.

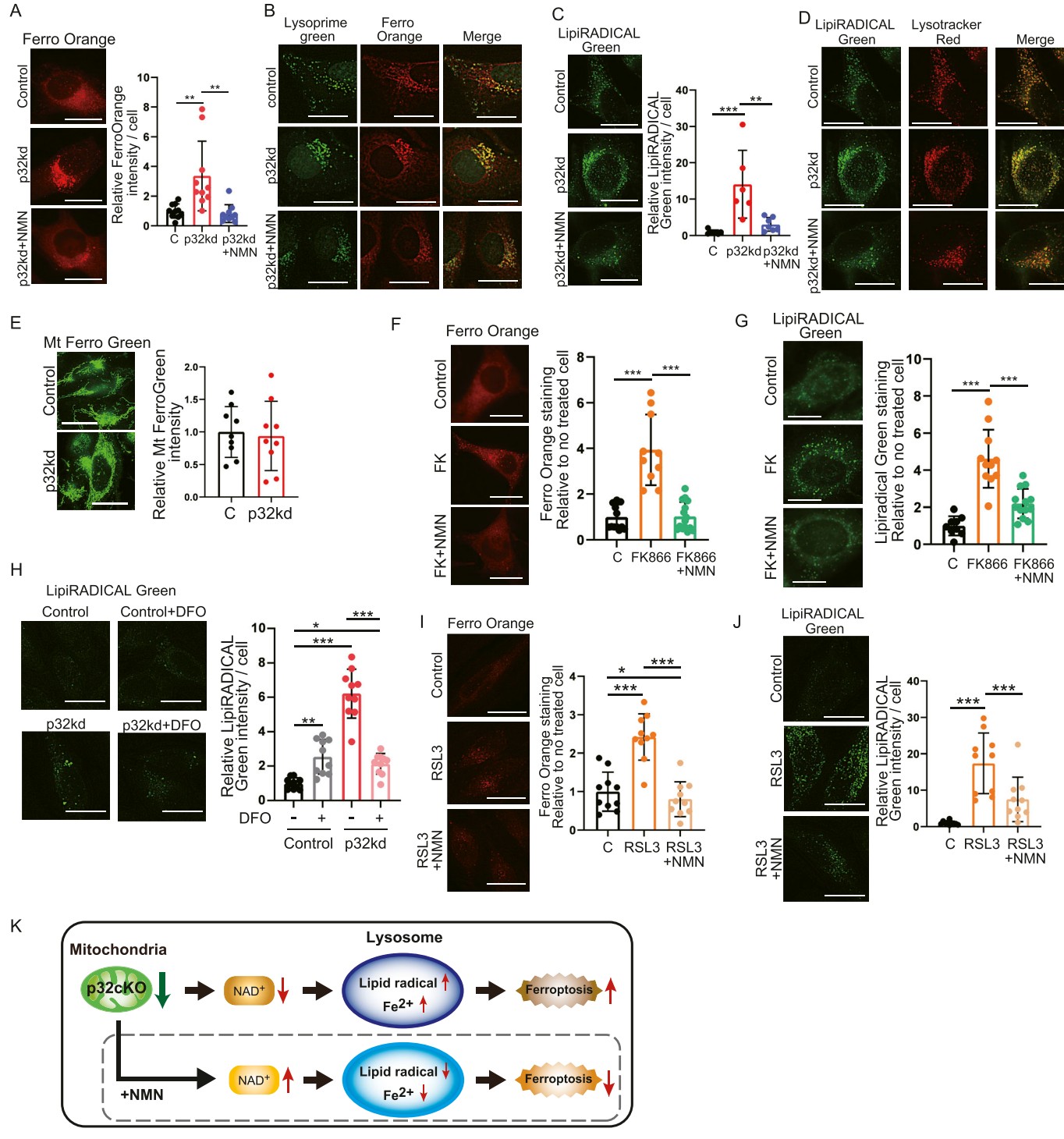

**Figure 7. Ferroptosis resulting from mitochondrial dysfunction is improved by nicotinamide mononucleotide (NMN).**
**(A)** Intracellular Fe$^{2+}$ staining by FerroOrange in HeLa cells. Control (C): non-targeting: p32kd: p32 knockdown, p32kd+NMN: p32 knockdown and addition of 1 mM NMN. The right panel shows the intensity of FerroOrange per cell (n = 10). **(B)** Intracellular Fe$^{2+}$ staining by FerroOrange and lysosomal staining by Lysoprime Green in HeLa cells. Lysosomal activity was reduced in p32 knockdown cells, and then intracellular Fe$^{2+}$ accumulated in lysosomes and yellow staining was increased. **(C)** Intracellular lipid-derived radical staining by LipiRADICAL Green in HeLa cells. The right panel shows the intensity of LipiRADICAL per cell (n = 6). **(D)** Intracellular lipid radical staining by LipiRADICAL Green and lysosomal staining by Lysotracker Red in HeLa cells. Lysosomal activity was reduced in p32 knockdown cells; therefore, LipiRADICAL Green accumulated in lysosomes and yellow staining was increased. **(E)** Mitochondrial Fe$^{2+}$ staining by MitoFerro Green in HeLa cells. The right panel shows the intensity of MitoFerro Green per cell (n = 10). **(F, G)** Staining by FerroOrange or LipiRADICAL Green in 3T3 cells. Control (C): untreated, FK866: addition of 10 nM FK866 for 48 h, FK866 + NMN: addition of 10 nM FK866 and 1 mM NMN for 48 h. Each intensity of FerroOrange and LipiRADICAL per cell (n = 10) is shown. **(H)** Staining by LipiRADICAL Green in HeLa cells. Control (C): untreated, deferoxamine (DFO): addition of 5 $\mu$M DFO for 48 h, p32kd: p32 knockdown, p32kd+DFO: p32 knockdown and addition of 5 $\mu$M DFO for 48 h. The

LipiRADICAL Green staining using deferoxamine (DFO), which is an iron chelator, as a ferroptosis inhibitor showed that the ferroptosis induced in p32 knockdown cells was improved by the addition of DFO (Fig 7H). This finding suggested that p32 was involved in inducing ferroptosis. Furthermore, ferroptosis was also induced with the GPX4 inhibitor RLS3, and this was ameliorated by the addition of NMN (Fig 7I and J). These results suggest that when mitochondrial translation is defective, $Fe^{2+}$ accumulates in lysosomes and lipid radicals are generated in lysosomes, inducing ferroptosis. The addition of NMN had an ameliorating effect on ferroptosis in vitro (Fig 7K).

## Discussion

Mitochondria are the main organelles for energy production, and mitochondrial dysfunction causes serious diseases associated with aging. We previously found that (1) the expression of $NAD^+$ synthase expression was decreased owing to mitochondrial dysfunction and (2) lysosomal and autophagic function was impaired owing to decreased $NAD^+$ biosynthesis (Yagi et al, 2021) as an additional cause of the exacerbation of chronic heart failure associated with aging. In this study, we found that myocardial markers, such as ANF, $\beta$-MHC, dopamine, and epinephrine, were increased in the myocardium of p32cKO mice. This finding indicates that these mice are a model for chronic heart failure. If cardiac function is improved by the administration of NMN, an $NAD^+$ precursor, it may prevent exacerbation of chronic heart failure and prolong a healthy lifespan. In the present study, we examined the preventive effect of NMN administration by reducing $NAD^+$ in the p32cKO model. We found that the ameliorative effect of NMN supplementation strongly affected lysosomal function, particularly lysosomal-mediated ferroptosis, rather than mitochondrial function.

In T-cell–specific, mitochondrial transcription factor A–deficient mice, mitochondrial $NAD^+$ levels were decreased and lysosomal function was impaired, which suppressed T-cell differentiation, but NAM (nicotinamide) administration restored $NAD^+$ levels (Baixauli et al, 2015). A decrease in $NAD^+$ levels was also reported in mice with aortic stenosis, which is a mouse model of acute cardiomyopathy (Diguet et al, 2018). In this model of acute cardiomyopathy, the expression of NAD nicotinamide riboside kinase 2 and NMN was increased, resulting in increased $NAD^+$ levels and the amelioration of acute cardiomyopathy (Diguet et al, 2018). Previous reports have also shown that NMN administration activates Sirt1 and restores cardiac function in ischemia–reperfusion models and in aging heart function (Yamamoto et al, 2014; Wei et al, 2021).

In the p32cKO model of chronic heart failure, $NAD^+$ levels are decreased with aging. Therefore, we administered NMN and investigated its restorative effect. We found that NMN not only had

anti-aging effects but also improved pathological conditions such as chronic heart failure. $NAD^+$-mediated mitochondrial and nuclear cross-talk and gene expression may be related (Gomes et al, 2013). In the p32cKO model in our study, a decrease in $NAD^+$ also induced expression of the nuclear transcription factor HIF-1$\alpha$. NMN administration then suppressed HIF-1$\alpha$ induction and improving $NAD^+$ levels also appeared to improve cross-talk between organelles.

NMN administration extends the lifespan of Ndufs4-KO mice, which is a mouse model of Leigh encephalopathy. However, the effect of NMN on an increase in $NAD^+$ in the brain could not be confirmed, and the mechanism of its lifespan-prolonging action has not yet been determined (Lee et al, 2019). In our p32cKO model of chronic heart failure, NMN administration improved lysosomal and autophagic function, but not mitochondrial function. We report that NMN improves chronic heart failure by improving NAD-mediated lysosomal function.

Mitochondria and lysosomes are associated with iron metabolism and are thought to interact with each other to regulate iron levels in vivo. Iron accumulates in mitochondria in a mouse model of ischemia–reperfusion or acute heart failure (Chang et al, 2016). Moreover, doxorubicin induces mitochondrial damage-dependent ferroptosis in a mouse model of acute heart failure, whereas Nrf2-mediated expression of HO1 has been reported to induce iron accumulation and ferroptosis (Fang et al, 2019). In our model of chronic heart failure, mRNA expression of the ferroptosis marker *Chac1* was also increased and ameliorated by NMN administration. GPX4 expression was decreased in a doxorubicin-induced mouse model of acute heart failure (Tadokoro et al, 2020), but in our chronic heart failure model, GPX4 protein levels were increased (Fig 6D). CHAC1-mediated GSH degradation functions as a promoter of ferroptosis. We observed a decrease in GSH levels with increased *Chac1* mRNA expression in the p32cKO model (Fig 6A) (Chen et al, 2021). Metabolite analysis showed that levels of GSH and GSSG, which are substrates of GPX4, were decreased, suggesting that GPX4 activity might be reduced, followed by a compensatory increase in *GPX4* gene expression (Fig 6D). Our results suggest that the pathway of ferroptosis differs between acute and chronic heart failure mouse models.

We previously found that p32/C1qbp-deficient mice, which have a mitochondrial translation defect, showed endoplasmic reticulum stress response and integrated stress response gene expression through ATF4 activation in the heart and brain (Sasaki et al, 2020). Interactions between CHOP and C/EBP$\beta$ were also reported to be required in mitochondrial cardiomyopathy to regulate the ATF4 regulatory transcriptional program (Kaspar et al, 2021). In mitochondrial myopathy muscle, mTORC1-activated ATF4 regulates the mitochondrial stress response (Khan et al, 2017). $Cox10^{-/-}$ mice develop early-onset lethal cardiomyopathy, which is associated with OXPHOS deficiency, lysosomal defects, and an aberrant mitochondrial morphology. The mice are associated with Oma1–

---

left panel shows the intensity of FerroOrange per cell (n = 10). **(I, J)** Staining by FerroOrange or LipiRADICAL Green in HeLa cells. Control (C): untreated, RLS3: addition of 0.5 $\mu$M RLS3 for 48 h, RLS3+NMN: addition of 0.5 $\mu$M RLS3 and 1 mM NMN for 48 h. Each intensity of FerroOrange and LipiRADICAL per cell (n = 10) is shown. **(A, B, C, D, E, F, G)** Data are presented as the mean ± SD. Scale bars, 20 $\mu$m. Data were analyzed by one-way analysis of variance with Tukey's multiple comparisons test. *$P$ < 0.05, **$P$ < 0.01, ***$P$ < 0.001. **(K)** Mechanism of mitochondrial translation deficiency. In p32cKO hearts, mitochondrial dysfunction led to reduced NAD synthesis, impaired lysosomal function, and iron accumulation in lysosomes. Treatment with NMN improved lysosomal function and recovery from ferroptosis.

Dele1–Atf4-dependent integrated stress response signaling and link ferroptosis in the heart (Ahola et al, 2022).

It was reported that mice with Cre expression from the αMHC promoter can show molecular and functional evidence of cardiotoxicity between 3 and 6 mo, possibly because of hidden loxP-sides in the mouse genome (Pugach et al, 2015). However, no marked gene expression such as ANF, βMHC, and FGF21 in p32$^{+/+}$: myh-cre mouse, suggesting that Cre overexpression does not affect in our mouse model at 6 mo.

Lysosomal dysfunction causes iron accumulation in lysosomes and induces ferroptosis (Torii et al, 2016). Because the induction of ferroptosis was confirmed in a mouse model of chronic heart failure, we hypothesized that abnormal mitochondrial and lysosomal function is responsible for abnormal iron metabolism. When p32 expression was decreased in a cultured cell system, lysosomal dysfunction, intracellular accumulation of $Fe^{2+}$, and induction of lipid radicals were observed. Furthermore, when intracellular NAD$^+$ levels were decreased by the Nampt inhibitor FK866, ferroptosis was induced, which suggested altered iron metabolism in lysosomes rather than in mitochondria. The finding that the addition of NMN reduced $Fe^{2+}$ accumulation and lipid radicals and improved ferroptosis suggests that NAD$^+$ is also important for iron metabolism.

Most intracellular iron is localized to mitochondria and lysosomes. Iron homeostasis is closely related to mitochondrial and lysosomal dynamics. Lysosomal dysfunction leads to abnormal iron metabolism and cell death by ferroptosis. However, this study shows that NMN administration improves lysosomal function and cell death by ferroptosis when NAD$^+$ levels are improved. In the near future, we aimed to apply NMN to clinical use in aging- and lysosome-related diseases.

# Materials and Methods

## Mouse models

Animal care was in compliance with Kyushu University Animal Care Guidelines (#A21-223). All experimental procedures conformed to the Guide for the Care and Use of Laboratory Animals, Eighth Edition, which were updated by the US National Research Council Committee in 2011. The animals were treated in accordance with the guidelines stipulated by Kyushu University Animal Care and Use Committee (Saito et al, 2017). Mice homozygous for the exon 3 floxed p32 allele (p32$^{flox/flox}$) were generated as described previously (Yagi et al, 2012). Mouse lines generated in this study have been deposited to the Knockout Mouse Project, B6.Cg-C1qbp<tm1Tuch>, and RBRC11610 (RIKEN). To obtain mice with heart-specific knockout of p32, we crossed p32$^{flox/flox}$ mice with αMHC-cre (Myh6 promoter) mice (Saito et al, 2017). NMN (Mirailab Bioscience Inc.) was dissolved in water at a concentration of 1 mg/ml and sterilized on a filter. NMN was mixed with drinking water from 2 mo of age, and analysis was performed at each month of age thereafter. Drinking water was changed once a week with the assumption that each mouse drank 5 ml of water each day. Only male mice were used in the experiments.

## Metabolomic assay

Heart-derived metabolites were analyzed by LC-MS/MS on the basis of reverse-phase ion-pair chromatography and hydrophilic interaction chromatography modes coupled with a triple quadrupole mass spectrometer (LCMS-8040; Shimadzu). To monitor metabolites, including intermediates in central metabolism, reverse-phase ion-pair chromatography was performed using an Acquity UPLC BEH C18 column (100 × 2.1 mm, 1.7-μm particle size; Waters). The mobile phase consisted of solvent A (15 mM acetic acid and 10 mM tributylamine) and solvent B (methanol), and the column oven temperature was 40°C. The gradient elution program was as follows: flow rate of 0.3 ml/min: 0–3 min, 0% B; 3–5 min, 0–40% B; 5–7 min, 40–100% B; 7–10 min, 100% B; and 10.1–14 min, 0% B. The parameters for the negative ESI mode under multiple reaction monitoring were as follows: drying gas flow rate, 15 liters/min; nebulizer gas flow rate, 3 liters/min; DL temperature, 250°C; heat block temperature, 400°C; and collision energy, 230 kPa. To monitor 61 types of metabolites including amino acids, HILIC chromatography was performed using a Luna 3u HILIC 200A column (150 × 2 mm, 3-μm particle size; Phenomenex). The mobile phase consisted of solvent A (10 mM ammonium formate in water) and solvent B (9:1 of acetonitrile:10 mM ammonium formate in water), and the column oven temperature was 40°C. The gradient elution program was as follows: flow rate of 0.3 ml/min: 0–2.5 min, 100% B; 2.5–4 min, 100–50% B; 4–7.5 min, 50–5% B; 7.5–10 min, 5% B; and 10.1–12.5 min, 100% B. The parameters for the positive and negative ESI modes under multiple reaction monitoring were as described above. Data processing was performed using the LabSolutions LC-MS software program (Shimadzu).

## RNA extraction

Immediately after dissection, the heart tissue was immersed in RNAlater (Invitrogen) overnight and then RNA was extracted. To perform RNA extraction from heart tissue, the ReliaPrep RNA Miniprep System (Promega) was used and the RNeasy Mini Kit (Qiagen) was used for cells. However, QIAshredder (Qiagen) was used before performing RNA extraction from cells. cDNA was synthesized using total RNA (500 ng), random hexamer primers, oligo dT primers, and the PrimeScript RT Reagent Kit (Takara).

## Real-time polymerase chain reaction (PCR) analysis

The cDNA was then subjected to real-time PCR analysis with TB Green Premix Ex Taq II (Takara) and the StepOnePlus Real-Time PCR System (Applied Biosystems). The expression level of each mRNA was normalized to the level of 18S ribosomal RNA obtained from the corresponding reverse transcription product. The primer sequences are listed in Table S1.

## Immunoblotting

Heart tissue was immediately frozen in liquid nitrogen. Tissues and cultured cells were lysed with lysis buffer (20 mM Tris–HCl, pH 7.5, 150 mM NaCl, 2 mM EDTA, 1% NP-40, 0.1% sodium dodecyl sulfate, and protease inhibitor cocktail [Wako]), homogenized by sonication, and

then subjected to immunoblotting. Protein was adjusted to 5 $\mu$g/lane using the BCA protein assay kit (Nacalai Tesque). Adjusted samples were separated using 8%, 10%, and 12% sodium dodecyl sulfate-polyacrylamide gel electrophoresis and transferred to PVDF membranes. The blocking buffer used was blocking one (Nacalai Tesque), and primary and secondary antibodies were diluted in Can Get Signal (Toyobo). All antibody dilutions were 1:5,000. Chemi-Lumi ImmunoStar LD (Fujifilm) was used for detection.

### Immunohistochemistry of heart sections

After mice were anesthetized with an overdose of sevoflurane, the hearts were harvested, and tissue sections were prepared from hearts fixed in 10% formaldehyde (Muto Pure Chemicals) to obtain paraffin-embedded coronal sections. These sections were then stained with various antibodies. Primary antibodies were diluted 1:200 in Can Get Signal (Toyobo). An argon laser light (488, 540 nm) was used to excite lipofuscin autofluorescence. A band-pass filter between 510 and 560 or 580 and 630 nm was used for fluorescence, and sections were observed under a microscope (BZ-X800; Keyence).

### Electron microscopy

To perform electron microscopy, samples were fixed with 2% PFA and 2% glutaraldehyde in 0.1 M phosphate buffer, pH 7.4, at 4°C overnight. After this fixation, washing was performed three times with 0.1 M phosphate buffer for 30 min each, followed by postfixing with 2% osmium tetroxide in 0.1 M phosphate buffer at 4°C for 2 h. Dehydration was carried out using graded ethanol solutions (50%, 70%, and 90%, anhydrous). The schedule was as follows: 50% and 70% for 30 min at RT, and four changes of anhydrous ethanol for 30 min each at RT. Infiltration with propylene oxide was performed twice for 30 min each, followed by immersion in a 70:30 mixture of propylene oxide and resin (Quetrol-812; Nisshin EM Co.) for 1 h. The cap of the tube was then kept open and propylene oxide was volatilized overnight. Samples were transferred to fresh 100% resin and polymerized at 60°C for 48 h. The polymerized resin was subjected to ultra-thin sectioning at 70 nm with a diamond knife using an ultramicrotome (Ultracut UCT; Leica), and the sections were mounted on copper grids. The sections were stained with 2% uranyl acetate for 15 min, washed with distilled water, and then secondarily stained with lead stain solution (Sigma-Aldrich) for 3 min. The grids were observed by a transmission electron microscope (JEM-1400Plus; Jeol) at an acceleration voltage of 100 kV. Digital images were captured using a CCD camera (EM-14830RUBY2; Jeol).

### Oxygen consumption rate measurement

Mitochondria were extracted from the heart. The heart tissue was crushed in isolation buffer (215 mM mannitol, 75 mM sucrose, 1 mM EDTA, and 20 mM HEPES, pH 7.4), sonicated, and centrifuged at 800$g$ for 10 min at 4°C to remove unbroken tissue and nuclei. The supernatant was further centrifuged at 9,100$g$ for 6 min at 4°C to enrich the mitochondria. The extracted mitochondria were adjusted to 1 $\mu$g/$\mu$l by the BCA protein assay and suspended in reaction buffer (215 mM mannitol, 75 mM sucrose, 1% BSA, 20 mM

HEPES, pH 7.4, 2 mM MgCl$_2$, and 2.5 mM KH$_2$PO$_4$). The oxygen consumption rate was measured using the Seahorse XFe24 Analyzer (Agilent) under basal conditions or following the addition of 1 mM ADP, 10 $\mu$M oligomycin, 4 $\mu$M FCCP (uncoupler), and 1 $\mu$M rotenone/antimycin A (electron transport inhibitor) in accordance with the manufacturer's protocol.

### Cell culture

All cell lines were cultured in DMEM (1,000 mg/liter glucose; Sigma-Aldrich) supplemented with 10% FBS at 37°C in a humidified atmosphere with 5% CO$_2$. The cells were incubated with the Nampt inhibitor FK866 (10 nM; Selleck Chemicals LLC) and/or 1 mM NMN for 48 h.

### Knockdown of p32

To perform p32 knockdown, short interfering RNA (Sigma-Aldrich) was transfected with Lipofectamine RNAiMAX (Thermo Fisher Scientific) in accordance with the manufacturer's instructions. In HeLa cells transfected after 24 h, 1 mM NMN (Oriental Yeast) was added, followed by culture for 48 h.

### Ferroptosis assay

Detection of intracellular Fe$^{2+}$ was performed using FerroOrange (excitation: 561 nm, emission: 570–620 nm) (Dojindo). Cells were cultured in glass-bottomed dishes for 3 d and washed with HBSS (Thermo Fisher Scientific), after which 1 $\mu$M FerroOrange was added for 30 min at 37°C, followed by observation under a microscope (BZ-X800; Keyence). Lipid radicals were detected by LipiRADICAL Green (Funakoshi) (excitation: 470 nm, emission: 520–600 nm). Cells were cultured in glass-bottomed dishes for 3 d and washed with HBSS, after which 1 $\mu$M LipiRADICAL Green was added for 10 min at 37°C, followed by observation under a microscope (BZ-X800; Keyence).

### Quantification and statistical analysis

The data are expressed as means ± SD of the indicated number of experiment and mice. Unpaired $t$ test was used to determine statistical differences between two groups. *$P < 0.05$, **$P < 0.01$, or ***$P < 0.005$ was considered statistically significant.

## Data Availability

Further information and requests for resources and reagents should be directed to and will be made available upon reasonable request by Takeshi Uchiumi (uchiumi.takeshi.008@m.kyushu-u.ac.jp). This study did not generate new reagents.

## Supplementary Information

# Acknowledgements

We thank Ellen Knapp, PhD, from Edanz (https://jp.edanz.com/ac) for editing a draft of this manuscript. We also appreciate technical assistance from the Research Support Center, Research Center for Human Disease Modeling, Kyushu University Graduate School of Medical Sciences. This work was supported by JSPS KAKENHI Grant Numbers JP22H03537, JP21K11678, JP20H00530, and JP17H01550.

## Author Contributions

M Yagi: conceptualization, resources, data curation, formal analysis, funding acquisition, validation, investigation, visualization, methodology, project administration, and writing—original draft, review, and editing.
Y Do: data curation, formal analysis, investigation, methodology, and writing—review and editing.
H Hirai: data curation, formal analysis, validation, investigation, methodology, and writing—review and editing.
K Miki: data curation, formal analysis, investigation, and writing—review and editing.
T Toshima: data curation, validation, investigation, and writing—review and editing.
Y Fukahori: data curation, formal analysis, investigation, methodology, and writing—review and editing.
D Setoyama: data curation, formal analysis, validation, methodology, and writing—review and editing.
C Abe: data curation, investigation, and writing—review and editing.
Y-I Nabeshima: data curation, formal analysis, supervision, investigation, methodology, and writing—review and editing.
D Kang: conceptualization, supervision, funding acquisition, and writing—review and editing.
T Uchiumi: conceptualization, resources, data curation, formal analysis, supervision, funding acquisition, validation, investigation, visualization, methodology, project administration, and writing—original draft, review, and editing.

## Conflict of Interest Statement

The authors declare that they have no conflict of interest.

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
