## [Reviewer comments · Life Science Alliance]

Life Science Alliance

Improving lysosomal ferroptosis with NMN administration protects against heart failure

Mikako Yagi, Yura Do, Haruka Hirai, Kenji Miki, Takahiro Toshima, Yukina Fukahori, Daiki Setoyama, Chiaki Abe, Yo-ichi Nabeshima, Dongchon Kang and Takeshi Uchiumi

DOI: <https://doi.org/10.26508/lsa.202302116>

Corresponding author(s): Dr. Takeshi Uchiumi (Kyushu University)

Review Timeline:

Submission Date:	2023-04-26
Editorial Decision:	2023-05-26
Revision Received:	2023-08-24
Editorial Decision:	2023-09-20
Revision Received:	2023-09-22
Accepted:	2023-09-25

Transaction Report:

May 26, 2023

Re: Life Science Alliance manuscript #LSA-2023-02116-T

Prof. Takeshi Uchiumi
Kyushu University
Department of Clinical Chemistry and Laboratory Medicine Kyushu University Graduate School of Medical Sciences
3-1-1 Maidashi, Higashi-ku
Fukuoka 812-8582
Japan

Dear Dr. Uchiumi,

Thank you for submitting your manuscript entitled "Improving lysosomal ferroptosis with NMN administration protects the heart failure" to Life Science Alliance. The manuscript was assessed by expert reviewers, whose comments are appended to this letter. We invite you to submit a revised manuscript addressing the Reviewer comments.

Thank you for this interesting contribution to Life Science Alliance. We are looking forward to receiving your revised manuscript.

Sincerely,

B. MANUSCRIPT ORGANIZATION AND FORMATTING:

Reviewer #1 (Comments to the Authors (Required)):

The study "Improving lysosomal ferroptosis with NMN administration protects the heart failure" by Yagi et al., reports that the administration of nicotinamide mononucleotide (NMN) is beneficial in an animal model of heart failure by inhibiting ferroptosis.

This paper is in direct continuation of the group's prior work (Yagi et al, 2021, Saito et al., 2017, Yagi et al., 2012). Many of the same mechanisms and findings related to NAD metabolism, HIF α regulation, and lysosomal function were reported in the earlier papers but in different tissues and cell systems. The connection to ferroptosis appears new.

The authors first reported on the role of p32 (official symbol C1QBP/C1qbp) in an elegant study in 2012 (Yagi et al., 2012, NAS).

In the current paper, they used mice with a conditional/heart-restricted p32 knock-out, which they described in more detail recently (Yagi 2021), to further study the potential of NMN supplementation.

The mechanisms of this and the prior studies are generally intriguing, and I enjoyed reading them.

I have the following comments or suggestions:

1. The authors conclude that the extended lifespan after NMN administration is due to improved cardiac function. They support this by changes in ANF and bMHC levels as well as heart tissue levels of dopamine and epinephrine. However, in this study they do not present any physiological or functional data. In Saito et al. 2017 (Cardiovascular research 113: 1173-1185) from the same group they provide basic echocardiographic data for these mice up to age 12 weeks. There, these mice have clear chamber dilatation in diastole and systole and deterioration of functional parameters (EF and FS). However, the EF in the p32KO mice is not dramatically reduced at 54% +/- 2% at 12 weeks (unchanged from 3 months). In the present study, the authors supplement NMN from 2 to 9 months. I am wondering if the EF/heart function declines post the 12-week mark, and maybe helps to explain the survival benefit.

2. Related to this, any additional objective data on heart function would very much help in understanding the effects, e.g., echocardiography data, exercise data, heart-to-body weight, lung weight changes, evidence of liver congestions or pulmonary vein congestion on histology to name a few options. Functional or imaging data, if not already collected, will be impossible to present basic vital signs and histological sections will hopefully be available to strengthen this statement.

3. Can the authors please clarify what the genotype of the control mice is? They are referred to p32wt here and in the Saito 2017 paper. But are these mice with only the p32flox/flox locus or are these not tamoxifen-treated p32flox/flox x aMHC-MerCreMer mice? Mice with Cre expression from the aMHC promoter can show molecular and functional evidence of cardiotoxicity between 3-6 months, possibly due to hidden loxP-sides in the mouse genome. These changes include increased fibrosis and DNA damage and are more severe in male animals (Pugach, JMCC, <http://dx.doi.org/10.1016/j.jmcc.2015.06.019>). All animals in the present study were male.
If the p32wt mice were "not tamoxifen-treated p32flox/flox x aMHC-MerCreMer mice", the authors could easily dismiss any unwanted side effects of the model/Cre expression as cause or confounder for some of the changes. Otherwise, they may want to include this possibility in the discussion. This is not to imply the basic findings and mechanisms are invalid or confounded, as some of them are supported in other tissues or cells.

4. The authors use ANOVA with Tukey's post-hoc throughout the study but most comparisons have $n = 5$ to ~ 12 . This is absolutely a reasonable number but a non-parametric test may (e.g., Kruskal-Wallis) may be more suitable in these cases?

5. The data is presented overall in a logical way, but I find the text sometimes hard to follow.

Reviewer #2 (Comments to the Authors (Required)):

The authors have previously shown that cardiac-specific p32cKO mice develop lethal cardiomyopathy and lysosomal dysfunction. Lysosomal dysfunction was shown to be linked to impaired NAD metabolism and was rescued by NMN supplementation in a cell culture model. In the present study, the authors further characterize the lysosomal defect and the likely link to disturbed mitophagy in the cardiac tissue. The beneficial effect of NMN treatment is well-documented. However, the link to ferroptosis and their interpretation of the role of the ISR are based on few observations only and appear to be premature. The schematic figure of the proposed mechanism in Figure 6G lacks sufficient experimental support, either in vivo or in vitro. Several inconsistencies related to Atf4 activation and ferroptosis regulation need to be clarified. The authors use only a few poorly characterized markers to draw rather far-reaching conclusions. I should add that the authors do not cite the current field in an adequate manner. For example, NAD replenish treatment has been tested in other mitochondrial mouse model as well and the Auwerx and Suomalainen labs have published on this topic extensively. Atf4 induction and integrated stress response in mitochondrial cardio-/myopathies has been shown by several labs, including the Trifunovic, Larsson and Suomalainen labs and the Langer lab has also linked Atf4 induction in a cardiomyopathy model to ferroptosis.

Major points:

- The characterization of the cardiac rescue is somehow superficial. Is the dilated cardiomyopathy in p32 KO rescued by NMN?
- To demonstrate ferroptosis in p32CKO mice heart, the authors must demonstrate lipid peroxidation (for example MDA or 4HNE staining) in the heart and assess various ferroptosis markers. Gpx4 accumulation is very convincing. How about other peroxidases and oxidases such as Txrnd1/ Txrnd2?
- Related to this point, how were marker proteins selected? Chac1 is an Atf4 target gene and participates in GSH degradation. Authors show increased expression of Atf4, and state that Chac1 expression is most likely due to Atf4 induction but still use it as ferroptosis marker?
- The authors state that Atf4 is induced by ER stress but do not provide evidence in support of this claim. Atf4 can be also induced by mitochondrial stress, lysosomal dysfunction, heme deprivation, viral infection etc. Are ER stress markers induced? Does stress induction depend on Oma1-Dele1-HRI, PERK or other kinases? What is the role of amino acid deprivation (considering the role of lysosomes for amino acid metabolism) and GCN2? They use Trib3 expression, as a read-out of ER stress, which however is also an Atf4 target.
- The authors state that Gpx4 protein expression is regulated by Chac1 but do not provide experimental evidence or refer to any publications where this has been shown. Gpx4 expression is regulated by Nrf2 and partially via Atf4. Their data may suggest that Nrf2 expression is reduced after NMN treatment, which could explain the reduced Gpx4 transcript and protein levels. Does NMN treatment affect ROS levels and thereby Nrf2?
- The authors use sodium selenite as a ferroptosis inducer, which is ambiguous and not well established. Selenium supports the synthesis of GSH as well as the synthesis of many anti-ferroptotic enzymes (such as Gpx4, thioredoxins etc) and thus can protect from ferroptosis. The authors should use erastin derivatives or the Gpx4 inhibitor RSL3 to induce ferroptosis and ferrostatin as a specific inhibitor for ferroptotic cell death. Does NMN suppress ferroptosis under these conditions? Is LipiRADICAL Green increased after erastin treatment?

Minor:

- The increase in NAD levels is very modest. Is NMN stable in the drinking water for 5days?
- Does Gpx4 localize to lysosomes/lysosomal surface upon ferroptosis?
- Reasoning for lipofuscin is very vague and should be better explained.
- "Protein expression results of LC3, another marker of autophagy, confirm the effect of NMN administration on autophagy (Figure S1D)". The results show however that the effect is not statistically significant.
- In Figures 4 and 5 bar charts with individual data points should be shown (as in other figures).
- The text would benefit from language editing.

Reviewer #1

Reviewers' Comments to Author Manuscript #LSA-2023-02116-T

We sincerely appreciate that you provided several critical comments. We are sure that your suggestions were very important for improvement of our manuscript. We added several experiments and extensively improve this revised manuscript according to your comments.

1. The authors conclude that the extended lifespan after NMN administration is due to improved cardiac function. They support this by changes in ANF and β MHC levels as well as heart tissue levels of dopamine and epinephrine. However, in this study they do not present any physiological or functional data. In Saito et al. 2017 (Cardiovascular research 113: 1173-1185) from the same group they provide basic echocardiographic data for these mice up to age 12 weeks. There, these mice have clear chamber dilatation in diastole and systole and deterioration of functional parameters (EF and FS). However, the EF in the p32KO mice is not dramatically reduced at 54% +/- 2% at 12 weeks (unchanged from 3 months).

In the present study, the authors supplement NMN from 2 to 9 months. I am wondering if the EF/heart function declines post the 12-week mark, and maybe helps to explain the survival benefit.

Thank you for reviewing our manuscript and for providing helpful comments.

Previously, we observed that p32cKO mice showed cardiac dysfunction at 2 months of age (n=8) and an 84% reduction in EF compared to control mice. In addition, the heart weight/body weight ratio was increased in p32cKO mice at 5 months of age compared to p32WT mice. (P32f/f: 73.8 ± 1.3 p32f/f:myh-cre: 62.7 ± 1.3 (**p<0.01))

In this study, gene expression of the heart failure markers ANF and β -MHC was also observed to be increased in p32cKO mice, which was partially ameliorated by NMN treatment (Fig. 1A). Analysis of dopamine and epinephrine metabolites also showed a significant increase in p32cKO myocardium, which was improved by NMN treatment (Fig. 1C). at 9 month old.

However, myocardial wall thickness in p32cKO mice were not improved by NMN

treatment (supplementary Fig S1A), suggesting that NMN treatment did not improve heart size or myocardial morphology. Basic echocardiographic analysis was not performed as it is a very sensitive method.

However, functional analysis, including analysis of ANF gene expression and metabolites (dopamine and epinephrine), confirmed that NMN treatment improved cardiac dysfunction in p32cKO hearts.

These results suggest that NMN treatment functionally ameliorated or prevented exacerbation of heart failure in p32 cardiomyocyte-specific knockout mice, thereby prolonging their lifespan.

In this revised manuscript, we added the sentence in Page 7 (Result section)

“However, NMN treatment did not improve the heart size or morphology (supplementary Fig S1A). These results suggest that NMN treatment functionally ameliorated or prevented exacerbation of heart failure in p32 cardiomyocyte-specific knockout mice, thereby prolonging their lifespan.”

2. Related to this, any additional objective data on heart function would very much help in understanding the effects, e.g., echocardiography data, exercise data, heart-to-body weight, lung weight changes, evidence of liver congestions or pulmonary vein congestion

on histology to name a few options. Functional or imaging data, if not already collected, will be impossible to present basic vital signs and histological sections will hopefully be available to strengthen this statement.

Thank you for reviewing our manuscript and for providing helpful comments.

Analyses of cardiac function, including echocardiographic data, exercise data, heart rate, lung weight changes, presence of liver congestion and pulmonary venous congestion, were not performed, as no differences in heart weight/body weight ratio or histological analyses were observed.

However, functional analysis, including analysis of ANF gene expression and

metabolites (dopamine and epinephrine), confirmed that NMN administration ameliorates cardiac dysfunction in p32cKO hearts. Basic echocardiographic analysis was not performed as it is a very sensitive method.

In this revised manuscript, we added the sentence in Page 7 (Result section)

“However, NMN treatment did not improve the heart size or morphology (supplementary Fig S1A). These findings suggest that NMN treatment functionally improves the myocardium of p32cKO mice, reducing heart failure and, ultimately, extending the lifespan.”

3. Can the authors please clarify what the genotype of the control mice is? They are referred to p32wt here and in the Saito 2017 paper. But are these mice with only the p32flox/flox locus or are these not tamoxifen-treated p32flox/flox x aMHC-MerCreMer mice? Mice with Cre expression from the aMHC promoter can show molecular and functional evidence of cardiotoxicity between 3-6 months, possibly due to hidden loxP-sides in the mouse genome. These changes include increased fibrosis and DNA damage and are more severe in male animals (Pugach, JMCC, <http://dx.doi.org/10.1016/j.yjmcc.2015.06.019>).

All animals in the present study were male.

Thank you for reviewing our manuscript and for providing helpful comments.

Control mice were p32flox/flox mice.

In previous experience (Saito et al 2017), we first checked the effect of Cre overexpression on p32^{+/+} and p32^{f/f}: mouse by using the gene expression. We observed no marked gene expression such as ANF, β MHC, and FGF21 in p32^{+/+}: myh-cre mouse, suggesting that Cre overexpression does not affect in our mouse model at 6 months.

Please reference to Saito’s paper or below (supplementary Figure 6B)

In this paper we compared p32flox/flox mouse with p32 (flox/flox) myh-cre mice because of no influence with cre overexpression.

In this revised manuscript, we added the sentence in Page 14~15

It was reported that Mice with Cre expression from the aMHC promoter can show

molecular and functional evidence of cardiotoxicity between 3-6 months, possibly due to hidden loxP-sides in the mouse genome (Pugach EK et al, 2015). However, no marked gene expression such as ANF, β MHC, and FGF21 in p32^{+/+}: myh-cre mouse, suggesting that Cre overexpression does not affect in our mouse model at 6 months.

Supplemental Figure 6 Expression of marked gene with Tamoxifen treatment and Cre expression in wild mouse. (A) The gene expression of p32^{WT} and p32^{CKO} heart at 6 weeks post-tamoxifen injection were analyzed by qRT-PCR. (B) Gene expression of p32^{+/+} and p32^{ff} heart crossed with myh-Cre mouse were analyzed by qRT-PCR. Data show the mean \pm SD of triplicate experiments and *p < 0.05; **p < 0.01; ***p < 0.005; vs. controls. (N=3 per each group)

If the p32^{wt} mice were "not tamoxifen-treated p32^{flox/flox} x aMHC-MerCreMer mice", the authors could easily dismiss any unwanted side effects of the model/Cre expression as cause or confounder for some of the changes.

Thank you for reviewing our manuscript and for providing helpful comments.

We agree that control mice were suitable for p32^{flox/flox} x aMHC-MerCreMer mice without tamoxifen, but we did not have and use such mice in this experiment. Thank you for your good advice for future experiences.

Otherwise, they may want to include this possibility in the discussion. This is not to imply the basic findings and mechanisms are invalid or confounded, as some of them are supported in other tissues or cells.

Thank you for reviewing our manuscript and for providing helpful comments.

In our mouse model, overexpression of Cre had no effect. thank you for the sharp findings of the reviewer, we will improve our future research.

4. The authors use ANOVA with Tukey's post-hoc throughout the study but most comparisons have $n = 5$ to ~ 12 .

This is absolutely a reasonable number but a non-parametric test may (e.g., Kruskal-Wallis) may be more suitable in these cases?

Thank you for reviewing our manuscript and for providing helpful comments.

In previous paper

“Long-Term Administration of Nicotinamide Mononucleotide Mitigates Age-Associated Physiological Decline in Mice” *Cell Metabolism* 24, 796-806 2016.

They used one-way ANOVA with Tukey's test, then we used this one-way ANOVA with Tukey's multiple comparisons test. However, multiple comparisons were performed for the mean of every other sample.

(<https://doi.org/10.1016/j.cmet.2016.09.013>, <https://doi.org/10.1016/j.redox.2019.101192>)

5. The data is presented overall in a logical way, but I find the text sometimes hard to follow.

Thank you for reviewing our manuscript and for providing helpful comments.

In this revised manuscript, we change the several sentence and text by native speaker.

Reviewer #2 :

Reviewers' Comments to Author

Manuscript #LSA-2023-02116-T

We are grateful for several critical comments. We are convinced that your suggestions were very important to improve our manuscript. We have added some experiments and have greatly improved this revised manuscript according to your comments.

1. The schematic figure of the proposed mechanism in Figure 6G lacks sufficient experimental support, either in vivo or in vitro. Several inconsistencies related to Atf4 activation and ferroptosis regulation need to be clarified. The authors use only a few poorly characterized markers to draw rather far-reaching conclusions.

Thank you for your helpful comment,

We change the Figure 6G in this revised manuscript. (change Figure 6J in this revised manuscript)

Previously we observed that

“When glioblastoma cells were exposed to a glucose-starved condition, mitochondrial translation inhibitor can cause mitochondrial dysfunction, activated the Nrf2 -Ho-1 pathway. After that, accumulation of lipid peroxides resulted from the accumulation of divalent iron, and cell death occurred via ferroptosis.” *Oncogenesis*, 2022 Vol 11 Page, 59-

These results suggested that mitochondria translation inhibitor induced Nrf2-Ho1 pathway, leading to accumulation of iron, then induced ferroptosis. In this study, we observed that Nrf2, and Ho1 gene expression were activated in p32cKO heart, then accumulate the iron and ferroptosis were induced. NMN administration were rescue this pathway. Then, we summarized this point in Fig 6J.

In another paper we observed that mitochondrial translation defect induced ATF4-CHOP1 activation and ATF4 directly induced CHOP1 expression by CHIP-PCR experience.

“Mitochondrial translation inhibition triggers ATF4 activation, leading to integrated

stress response but not to mitochondrial unfolded protein response” *Bioscience reports* 2020 vol 40

In this study, it was also observed that ATF4 and CHOP gene expression was induced in p32cKO hearts and that NMN administration rescued this pathway.

Therefore, the schema in Fig. 6J was added.

CHAC1 is regulated by the ATF4 transcription factor and the function of ChaC1 is to degrade glutathione, which in turn reduces glutathione (GSH) levels. Since GSH is a substrate for Gpx4, reduced GSH levels lead to depletion of GPX4 activity and subsequent ferroptosis

“Lipid Peroxides Mediated Ferroptosis in Electromagnetic Pulse-Induced Hippocampal Neuronal Damage via Inhibition of GSH/GPX4 Axis” *International Journal of Molecular Sciences* 2022 16

In this study, we observed that Chac1 induction, reduced GSH, leading to ferroptosis in p32cKO heart, NMN administration rescue this pathway, then we summarized in Fig 6J in this revised manuscript.

2. I should add that the authors do not cite the current field in an adequate manner. For example, NAD replenish treatment has been tested in other mitochondrial mouse model as well and the Auwerx and Suomalainen labs have published on this topic extensively. Atf4 induction and integrated stress response in mitochondrial cardio-/myopathies has been shown by several labs, including the Trifunovic, Larsson and Suomalainen labs and the Langer lab has also linked Atf4 induction in a cardiomyopathy model to ferroptosis.

In this revised manuscript, we cited the several paper in Introduction
Introduction (Page 5)

“Nicotinamide phosphoribosyl transferase (NAMPT) plays a major role in the adipose-to-brain signaling pathway that sustains the association between increased adiposity and impaired sociability triggered by early life stress. In adipose

tissue-specific normalization of NAMPT levels, dietary treatment with NMN in adolescent stressed mice normalizes the alterations in sociability and neuronal excitability through the activation of the NAD⁺/sirtuin 1 (SIRT1) pathway (Morató *et al*, 2022). Furthermore, oral supplementation with nicotinamide riboside, a vitamin B3 form and NAD⁺ precursor, efficiently prevents the development and progression of mitochondrial myopathy in mice (Khan *et al*, 2014).”

In Discussion section (Page 14)

We added the sentence

“We previously found that p32/C1qbp-deficient mice, which have a mitochondrial translation defect, showed endoplasmic reticulum stress response and integrated stress response gene expression through ATF4 activation in the heart and brain (Sasaki *et al*, 2020). Interactions between CHOP and C/EBPβ were also reported to be required in mitochondrial cardiomyopathy to regulate the ATF4 regulatory transcriptional program (Kaspar *et al*, 2021). In mitochondrial myopathy muscle, mTORC1-activated ATF4 regulates the mitochondrial stress response (Khan *et al*, 2017). Cox10^{-/-} mice develop early-onset lethal cardiomyopathy, which is associated with OXPHOS deficiency, lysosomal defects, and an aberrant mitochondrial morphology. The mice are associated with Oma1-Dele1-Atf4-dependent integrated stress response signaling and link ferroptosis in the heart. (Ahola *et al*, 2022).

Major points:

1. The characterization of the cardiac rescue is somehow superficial. Is the dilated cardiomyopathy in p32 KO rescued by NMN?

Thank you for reviewing our manuscript and for providing helpful comments.

According to our previous report, p32cKO mice exhibited cardiac dysfunction at 2 months of age (n = 8), and we observed an 80% reduction in EF by echocardiography. Moreover, the heart weight/body weight ratio of 5-month-old mice was increased in p32cKO mice compared with p32WT mice. However, NMN administration did not improve heart weight/body weight ratio and myocardial wall thickness in p32cKO mice (Supplementary Fig. S1A). This suggests that NMN administration did not improve heart size or myocardial morphology.

However, NMN treatment improved cardiac function. In this study, we observed that

NMN treatment improved the gene expression of heart failure markers ANF and β -MHC in p32cKO mice (Fig. 1A). Furthermore, analysis of dopamine and epinephrine metabolites was also improved by NMN treatment (Figure 1C). These results suggest that NMN administration functionally ameliorates heart failure and prolongs life span in p32 cardiomyocyte-specific knockout mice.

2. To demonstrate ferroptosis in p32CKO mice heart, the authors must demonstrate lipid peroxidation (for example MDA or 4HNE staining) in the heart and assess various ferroptosis markers. Gpx4 accumulation is very convincing. How about other peroxidases and oxidases such as Txrd1/ Txrd2?

Thank you for reviewing our manuscript and for providing helpful comments.

Additional experiments with lipid peroxides such as 4-HNE and 3NT showed that lipid peroxidation was increased in p32cKO hearts, leading to ferroptosis.

We added the Figure in Figure 6H, 6I and supplementary Fig S3B, S3C in this revised manuscript.

We added the sentence in this revised manuscript, (Page 11)

“Lipid peroxide (4-HNE, 3-nitrotyrosine) in the heart was examined. Protein levels and tissue staining showed that lipid peroxide was accumulated intracellularly in p32cKO mice and improved by NMN administration in all experiments (Fig 6H, 6I and supplementary Fig S3B, S3C). The ferroptosis-related factors *Chac1*, GPX4, and *Hol* were also ameliorated by NMN (Fig 6J), which suggested that ferroptosis in the p32cKO heart was improved by NMN administration.

We also examined whether the mRNA expression of the Txrd1 and Txrd2 gene was changed, but we found no significantly changed. Protein expression was confirmed for TXNRD1, but no improvement was observed with the addition of NMN.

Real-time PCR analysis of RNA expression in the heart of WT and p32cKO mice treated with NMN from 2 to 11 months of age (WT, p32cKO, and p32cKO + NMN: n = 5). *Txnrd1* and *Txnrd2* were used as other peroxidases and oxidases. Data are presented as the mean \pm SD. Data were analyzed by one-way analysis of variance with Tukey's multiple comparisons test. * $p < 0.05$, ** $p < 0.01$, *** $p < 0.001$.

Western blot analysis of TXNRD1 in the heart of 11-month-old WT and p32cKO mice, and of p32cKO mice treated with NMN from 2 to 11 months of age (WT and p32cKO + NMN: n = 4, p32cKO: n = 5). GAPDH was used as an internal control. Quantification is shown in a graph.

In this revised manuscript, we did not show these results.

3. Related to this point, how were marker proteins selected? Chac1 is an Atf4 target gene and participates in GSH degradation. Authors show increased expression of Atf4, and state that Chac1 expression is most likely due to Atf4 induction but still use it as ferroptosis marker?

Thank you for reviewing our manuscript and for providing helpful comments.

Referring to the paper, gene expression of the ferroptosis markers,

prostaglandin-endoperoxide synthase 2 (*Ptgs2*), acyl CoA-synthetase long-chain family member 4 (*Acsl4*), and ChaC glutathione specific gamma-glutamylcyclotransferase 1 (*Chac1*) (Chen *et al.*, 2021). CHOP is an endoplasmic reticulum stress marker and also thought to be involved in ATF4-CHOP-CHAC1 pathway-induced ferroptosis (Wang *et al.*, 2019) (Mungrue *et al.*, 2009). We believe that ATF4 regulates the expression of CHOP and CHAC1. (page 10)

Referring to the paper, gene expression of the ferroptosis markers *Acsl4*, *PTGS2* and *CHAC1* was examined and found that *CHAC1* expression was upregulated in p32cKO and recovered with the addition of NMN; ATF4 is a known transcription factor for *CHAC1* and ER stress is known to expression and induces ferroptosis.

<https://doi.org/10.4049/jimmunol.182.1.466>

<https://doi.org/10.1016/j.bbrc.2019.09.023>

In this study, we observed that in P32cKO cardiac tissue, ATF4, CHAC1 and CHOP are up-regulated and restored by NMN. This may be a consequence of induced ferroptosis. In this revised manuscript, we described that mitochondrial stress induces the ATF4-CHAC1 pathway, leading to ferroptosis; NMN administration rescues these pathways and improves heart failure (Figure 6J) .

4. The authors state that *Atf4* is induced by ER stress but do not provide evidence in support of this claim. *Atf4* can be also induced by mitochondrial stress, lysosomal dysfunction, heme deprivation, viral infection etc. Are ER stress markers induced? Does stress induction depend on *Oma1-Dele1-HRI*, *PERK* or other kinases? What is the role of amino acid deprivation (considering the role of lysosomes for amino acid metabolism) and GCN2? They use *Trib3* expression, as a read-out of ER stress, which however is also an *Atf4* target.

Thank you for reviewing our manuscript and for providing helpful comments.

Previous reports have shown increased mitochondrial stress and transcription factor *ATF4* expression in p32cKO mice lead to expression of ER stress genes such as *CHOP1*, *Trib3* (Saito *et al.*, 2017; Sasaki *et al.*, 2017; Sasaki *et al.*, 2020a). Therefore, we assume that mitochondrial dysfunction alters *ATF4* expression. Among the ER stress markers, only gene expression of *CHOP1* was improved by NMN (Figure 6 C). Therefore, no further

experiments involving ER stress were performed. Gene expression of GCN2 was unchanged in p32cKO compared to WT.

In this study, we focus the ferroptosis in lysosome in p32cKO heart, then we toned down the ATF4 expression by ER stress and delete the ER stress in this revised manuscript.

5. The authors state that Gpx4 protein expression is regulated by Chac1 but do not provide experimental evidence or refer to any publications where this has been shown. Gpx4 expression is regulated by Nrf2 and partially via Atf4. Their data may suggest that Nrf2 expression is reduced after NMN treatment, which could explain the reduced Gpx4 transcript and protein levels. Does NMN treatment affect ROS levels and thereby Nrf2?

Thank you for reviewing our manuscript and for providing helpful comments.

Sorry for the confusion, but lipid peroxidation regulated by Gpx4 activation is dependent on GSH substrate levels. And since CHAC1 degrades GSH, I stated that Gpx4 activity is involved in CHAC1 expression. CHAC1-mediated GSH degradation is known to function as a promoter of ferroptosis, and we observed a decrease in GSH with increased CHAC1 expression in p32cKO model and an ameliorative effect by NMN was also observed (Figure. 6A). Nrf2 gene expression was increased in p32cKO mice and improved by NMN addition. Thus, ATF4, Nrf2, CHAC1, and GPX4 all show the same behavior of decreased function in p32cKO and improved function with NMN, suggesting a correlation, but this cannot be ruled out. In this revised manuscript, several sentences have been changed (Figure. 6) (page10-11).

In this revised manuscript, several sentences have been changed.

6. The authors use sodium selenite as a ferroptosis inducer, which is ambiguous and not well established. Selenium supports the synthesis of GSH as well as the synthesis of many anti-ferroptotic enzymes (such as Gpx4, thioredoxins etc) and thus can protect from ferroptosis. The authors should use erastin derivatives or the Gpx4 inhibitor RSL3 to induce ferroptosis and ferrostatin as a specific inhibitor for ferroptotic cell death.

Does NMN suppress ferroptosis under these conditions? Is LipiRADICAL Green increased after erastin treatment?

Thank you for reviewing our manuscript and for providing helpful comments.

According to the review's comment, we used erastin and RSL3 which induced ferroptosis.

We observed that the treatment of erastin which induced ferroptosis also increased the intensity of cytosolic iron ions, but decreased by addition of NMN (Figure S4A).

We also observed that the treatment of erastin which induced ferroptosis also increased the intensity of lipid peroxide using the LipiRADICAL Green, but decreased by addition of NMN (Figure S4B).

It was also observed that the Gpx4 inhibitor RSL3 increased cytosolic iron ions, leading to increased lipid peroxidation, and that NMN treatment inhibited the ferroptosis pathway (Figure 7I, 7J).

Furthermore, we used the iron chelator DFO (deferoxamine) as a ferroptosis inhibitor, and lipid radical green staining showed that the ferroptosis induced in p32 knockdown cells was improved by the addition of DFO. These suggests that p32 is involved in the induction of ferroptosis. (Figure 7H)

In this revised manuscript, we added the sentence in Result section and change the Figure 7, supplementary Figure S3, Figure S4.

Minor:

1. The increase in NAD levels is very modest. Is NMN stable in the drinking water for 5 days?

Thank you for review's comment, we considered that Mass spectrometry of the diluted NMN proved that the NMN was stable. As NMN administered to mice is replaced in a week, the purity of NMN was checked on day 1 and day 5 after NMN dilution: 100% on day 1, but 95.9% on day 5. NMN was considered stable.

2. Does Gpx4 localize to lysosomes/lysosomal surface upon ferroptosis?

Thank you for review's comment.

We are in the process of confirming that GPX4 localizes to lysosomes when ferroptosis is induced. We have also found that GPX4, which was co-immunoprecipitated with the

lysosomal membrane protein TMEM192, partially localized to lysosomes. We are currently preparing another paper using these data, which is not included in this paper.

Immunoprecipitation of samples using HA-TMEM192-transfected cells in U87 cells.

3. Reasoning for lipofuscin is very vague and should be better explained.

Recently, it was reported that lipofuscin is an insoluble fluorescent pigment that autofluoresces from lipid peroxides that accumulate with age in the lysosomal compartment. Thus, as lysosomal function declines, lipofuscin accumulates within lysosomes and can cause disease (Pierzynowska *et al*, 2021).

Therefore, we considered the accumulation of lipofuscin to be one indicator of impaired lysosomal function.

Then, in this revised manuscript, we change the sentence in Page 8

“Lipofuscin is an insoluble fluorescent dye that autofluoresces from lipid peroxides that accumulate in the lysosomal compartment with age. Therefore, as lysosome become less functional, lipofuscin may accumulate within lysosomes and cause disease.”

4. "Protein expression results of LC3, another marker of autophagy, confirm the effect of NMN administration on autophagy (Figure S1D)". The results show however that the effect is not statistically significant.

According to the review's comment, we deleted the supplementary Figure S1D in this revised manuscript. And we deleted the sentence in Page 8 in Result section

“Protein expression results of LC3, another marker of autophagy, confirm the effect of NMN administration on autophagy (Figure S1D).”

5. In Figures 4 and 5 bar charts with individual data points should be shown (as in other figures).

According to the review' s comment, we have changed the Figure 4 and Figure 5 as individual dot blots in this revised manuscript.

6. The text would benefit from language editing.

According to the review' s comment, we corrected the English by native speakers in this revised manuscript.

September 20, 2023

RE: Life Science Alliance Manuscript #LSA-2023-02116-TR

Prof. Takeshi Uchiumi
Kyushu University
Department of Clinical Chemistry and Laboratory Medicine Kyushu University Graduate School of Medical Sciences
3-1-1 Maidashi, Higashi-ku
Fukuoka 812-8582
Japan

Dear Dr. Uchiumi,

Thank you for submitting your revised manuscript entitled "Improving lysosomal ferroptosis with NMN administration protects the heart failure". We would be happy to publish your paper in Life Science Alliance pending final revisions necessary to meet our formatting guidelines.

- please address Reviewer 2's remaining minor comments
- please upload your Tables in editable .doc or excel format
- please upload all figure files as individual ones, including the supplementary figure files
- please add ORCID ID for the corresponding author -- you should have received instructions on how to do so
- please add the Twitter handle of your host institute/organization as well as your own or/and one of the authors in our system
- please note that titles in the system and on the manuscript file must match
- please add a callout for Figure S2B to your main manuscript text

A. FINAL FILES:

B. MANUSCRIPT ORGANIZATION AND FORMATTING:

Sincerely,

Reviewer #1 (Comments to the Authors (Required)):

The authors addressed my questions sufficiently. I do not have any further comments suggestions.

Reviewer #2 (Comments to the Authors (Required)):

In the revised version, the authors have addressed the points requested by the reviewers and have strengthened the manuscript. The data concerning in vitro ferroptotic assays is extensive and convincing. Although different interpretations for the causal relationship of some of their observations remain possible, the authors use careful and adequate wording to describe the findings. Only some minor textual corrections are necessary:

1. P. 6 "The expression of the ferroptosis-related genes prostaglandin-endoperoxide synthase 2 (Ptgs2), acyl CoA-synthetase long-chain family member 4 (Acsl4), and ChaC glutathione specific gamma-glutamylcyclotransferase 1 (Chac1) (Chen X et al, 2021, Mungrue IN et al, 2009, Wang N et al, 2019)."...a verb is missing in this sentence.
2. Figure 7B has a typo "Ferro Ogange"

We sincerely appreciate that you provided several critical comments. We are sure that your suggestions were very important for improvement of our manuscript. We added several experiments and extensively improve this revised manuscript according to your comments.

Reviewer #2 (Comments to the Authors (Required)):

In the revised version, the authors have addressed the points requested by the reviewers and have strengthened the manuscript. The data concerning in vitro ferroptotic assays is extensive and convincing. Although different interpretations for the causal relationship of some of their observations remain possible, the authors use careful and adequate wording to describe the findings. Only some minor textual corrections are necessary:

1. P. 6 "The expression of the ferroptosis-related genes prostaglandin-endoperoxide synthase 2 (*Ptgs2*), acyl CoA-synthetase long-chain family member 4 (*Acs14*), and ChaC glutathione specific gamma-glutamylcyclotransferase 1 (*Chac1*) (Chen X et al, 2021, Mungrue IN et al, 2009, Wang N et al, 2019)."...a verb is missing in this sentence.

Thank you for your comment

We changed the sentence in this re-revised manuscript

In Page 6

"The expression of the ferroptosis-related genes prostaglandin-endoperoxide synthase 2 (*Ptgs2*), acyl CoA-synthetase long-chain family member 4 (*Acs14*), and ChaC glutathione specific gamma-glutamylcyclotransferase 1 (*Chac1*) has been reported to be upregulated by ferroptosis (Chen X et al, 2021, Mungrue IN et al, 2009, Wang N et al, 2019)."

2. Figure 7B has a typo "Ferro Ogange"

Thank you for your comment, we changed the Typing "Ferro Orange" in this re-revised manuscript in Figure 7B .

September 25, 2023

RE: Life Science Alliance Manuscript #LSA-2023-02116-TRR

Dr. Takeshi Uchiumi
Kyushu University
Clinical Chemistry and Laboratory Medicine
3-1-1 Maidashi
Fukuoka, Fukuoka 812-8582
Japan

Dear Dr. Uchiumi,

Thank you for submitting your Research Article entitled "Improving lysosomal ferroptosis with NMN administration protects against heart failure". It is a pleasure to let you know that your manuscript is now accepted for publication in Life Science Alliance. Congratulations on this interesting work.

DISTRIBUTION OF MATERIALS:

Again, congratulations on a very nice paper. I hope you found the review process to be constructive and are pleased with how the manuscript was handled editorially. We look forward to future exciting submissions from your lab.

Sincerely,
